# An Overview of the Potential Medicinal and Pharmaceutical Properties of Ru(II)/(III) Complexes

**DOI:** 10.3390/ijms24119512

**Published:** 2023-05-30

**Authors:** Anna Skoczynska, Andrzej Lewinski, Mateusz Pokora, Piotr Paneth, Elzbieta Budzisz

**Affiliations:** 1Department of Endocrinology and Metabolic Diseases, Medical University of Lodz, 93-338 Lodz, Poland or anna_sko@onet.pl (A.S.); andrzej.lewinski@umed.lodz.pl (A.L.); 2International Center of Research on Innovative Biobased Materials (ICRI-BioM)—International Research Agenda, Lodz University of Technology, Zeromskiego 116, 90-924 Lodz, Polandpiotr.paneth@p.lodz.pl (P.P.); 3Institute of Applied Radiation Chemistry, Lodz University of Technology, Zeromskiego 116, 90-924 Lodz, Poland; 4Department of the Chemistry of Cosmetic Raw Materials, Medical University of Lodz, 90-151 Lodz, Poland

**Keywords:** coordination complexes, antitumor activity, type 2 diabetes, HIV, Alzheimer’s disease, molecular docking, photosensitizers, clinical trials

## Abstract

This review examines the existing knowledge about Ru(II)/(III) ion complexes with a potential application in medicine or pharmacy, which may offer greater potential in cancer chemotherapy than Pt(II) complexes, which are known to cause many side effects. Hence, much attention has been paid to research on cancer cell lines and clinical trials have been undertaken on ruthenium complexes. In addition to their antitumor activity, ruthenium complexes are under evaluation for other diseases, such as type 2 diabetes, Alzheimer’s disease and HIV. Attempts are also being made to evaluate ruthenium complexes as potential photosensitizers with polypyridine ligands for use in cancer chemotherapy. The review also briefly examines theoretical approaches to studying the interactions of Ru(II)/Ru(III) complexes with biological receptors, which can facilitate the rational design of ruthenium-based drugs.

## 1. Introduction

Coordination chemistry is a field of scientific research that focuses on various areas of life but mainly medicine and pharmacy. Rosenberg discovered cisplatin in 1965, a milestone for metallopharmaceutical development in anticancer therapy [1]. Since then, other Pt(II) compounds, such as **2** and **3** (Figure 1), have been developed for cancer treatment. Cisplatin (**1**) and other platinum complexes are effective against cancer of the neck, ovary, head, cervix, bladder and lymphoma. Despite their anticancer effectiveness, they have some disadvantages, such as toxicity to the excretory and nervous systems, as well as loss of hair, nausea, suppression of bone marrow, vomiting and resistance to drugs [1,2,3,4,5].

As many therapies fail due to the development of resistance of tumors to platinum compounds, there is great interest in identifying alternative anticancer drugs with mild side effects. Much hope is placed on complexes of Ru(II)/(III) ions, which are candidates for the next generation of metallopharmaceuticals in the treatment of cancer [1,6,7,8], because they have six-coordinated octahedron geometry and different degrees of oxidation drugs [1,2,9,10], which make them more attractive than Pt(II) analogues and allow easier ligand modifications. The low kinetics of ligand exchange enables the attachment of Ru(II/III) complexes to certain cell structures throughout the cell cycle. Ru(III) ions can exploit the transferrin overexpression in the cancer cells by mimicking Fe(III) ions and binding to the protein. This supports the selectivity of ruthenium complexes when entering neoplastic cells [1,2,11,12,13]. 

**Figure 1 ijms-24-09512-f001:**
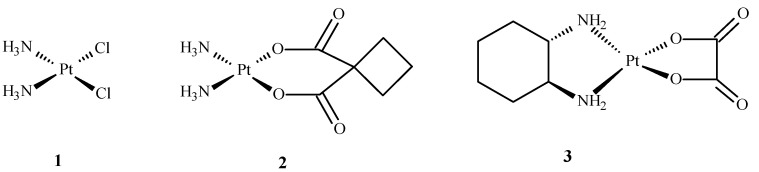
Structures of Pt(II) complexes: cisplatin (**1**), carboplatin (**2**), oxaliplatin (**3**) [10].

Ruthenium ions are taken into account [1,7,8,9,10] because their mechanism of action differs from platinum(II) [1,14,15,16,17]. As metal ions and metals are needed to maintain biological processes, they are often used in the medical diagnostics and treatment [1,18,19]. Ruthenium complexes represent a potential new generation of anti-tumor drugs and have many advantages over platinum complexes, such as avoiding drug resistance and lower toxicity.

Ruthenium occurs in the II, III, IV, VI and VIII oxidation states. Complexes of Ru(III) are less kinetically and thermodynamically more stable than complexes of Ru(II) [1,20]. They can be used as prodrugs in the cancer cells, where there is a lack of oxygen, high levels of glutathione and acidic pH. These factors favor changes in the oxidation state of Ru(III) ions, which are reduced to the Ru(II) ions. No Ru(II)/(III) compounds with anti-tumor activity have been commercialized.

Ever since compound **1** was subjected to clinical trials in 1979, more work has been put into the development of metallopharmaceuticals. Many metal complexes have anti-tumor potential, and ruthenium complexes are designed to mimic platinum complexes in targeting DNA. The complexes of Pt(II) form adducts with DNA based on cross-linkages, and that is their characteristic mechanism of DNA degradation in the cancer cells. As was mentioned before, ruthenium compounds have an octahedron shape, unlike the flat-square geometry of platinum compounds, and this influences the way they function and their activity mechanisms. The ability of ruthenium complexes to interact with DNA varies, and this is reflected in their cytotoxicity: some compounds of Ru(III) interact with nuclear DNA, decrease RNA synthesis, inhibit DNA replication and have mutagenic activity.

## 2. Main Mechanisms of Action of Ru(II)/(III) Complexes and Their Therapeutic Targets

Ruthenium complexes have multiple molecular targets and various mechanisms of anti-tumor activity. In Figure 2, molecular targets that are important in fighting cancer cells are shown.

Some complexes of Ru(II) ions interact with telomeric DNA, but others affect DNA replication and transcription. Referring to the literature data, some complexes of Ru(III) are prodrugs because they are reduced to Ru(II) ions; these observations have resulted in the performance of studies on the anti-tumor properties of Ru(II) compounds containing an arene substituent [21]. Studies of [Ru(η^6^-C_6_H_6_)(DMSO)Cl_2_] (**4**), the first arene Ru(II) compound (Figure 3), revealed that it strongly inhibits topoisomerase II, which participates in DNA replication and is responsible for structural organization [21]. Scolaro et al. [21,22] showed that replacing dimethyl sulfoxide (DMSO) with pta ligand, known as 1,3,5-triaza-7-phosphatricyclo [3.3.1.1]decane, caused an increase in RAPTA-C (**5**) water solubility. Other complexes, KP1558 and its analog containing 3,5,6-bicyclophosphine-α-D-glucofuranoside (**6**) instead of pta (Figure 3), showed high anti-tumor activity [21]. The transformation of the carbohydrate part of the compounds **7**–**9** (Figure 3) increased their water solubility, and all complexes exhibited high cytotoxicity towards human ovarian cancer (CH1), colorectal cancer (SW480), lung cancer (A549), melanoma proteoglycan cells (Me300), glioblastoma (LNZ308) and cisplatin resistant ovarian cancer (A2780cisR). Moreover, it is believed that the alkyl part of phosphine enhanced the cytotoxicity and strengthened the lipophilicity [21]. In general, complex **9** (Figure 3) is the most active and complex **7** is the least active (Figure 3).

Colorectal cancer (SW480), lung cancer (A549) and ovarian cancer (CH1) cell lines were exposed to compounds **10**–**15** (Figure 4); they were found to be most active against SW480 cells and CH1 cells. One experiment used transgenic mouse cells that contained human colonic adenocarcinoma cells (LS174T). The mice were administered RAPTA-C (**5**) at 100 mg/kg for eleven days. The tumor growth was found to be reduced by 50% compared to the control, while doses of 10 and 40 mg/kg did not inhibit tumor growth. Moreover, ruthenium complexes can block the cell cycle [21,23,24,25] and induce cross-linking with DNA, which leads to cell apoptosis [21,26,27]. Complexes of Ru(II) with polypyridine stably bind to quadruplex DNA (G4-DNA) in the telomeric structure of DNA [28,29], thus, blocking DNA replication and preventing the growth of cancer cells [30,31]. In general, ruthenium compounds show good topoisomerase inhibitory activity and can accumulate in the mitochondria, the endoplasmic reticulum, lysosomes and the cell nucleus [32,33]. The mitochondria seem to be the primary therapeutic targets for ruthenium compounds [34,35,36] as treatment lowers the mitochondrial membrane potential, thus, leading to its dysfunction or apoptosis. Moreover, they lead to the promotion of the release of Bcl-2 family proteins, the release of cytochrome c and caspase cascade activation, resulting in the death of neoplastic cells. Ruthenium complexes may find their way into the endoplasmic reticulum disease, inducing oxidative stress and inducing the apoptosis of neoplastic cells by activating caspases [37,38].

In addition, Ru(II)/(III) complexes can attack lysosomes by inducing autolysosome production and the hydrolase release [39,40,41]. Thus, they increase the apoptosis of neoplastic cells [42].

An important feature of the described compounds is their efficacy against many platinum-resistant neoplasms. Zhao and Chen [43] designed Ru(II) polypyridine complexes targeting the mitochondria, based on 2-phenylimidazo [4,5-f][1,10]phenanthroline (PIP). These compounds induced apoptosis by the mitochondrial pathway and showed effectiveness against cisplatin—resistant A2780cisR tumor cells [34,44,45].

**Figure 4 ijms-24-09512-f004:**
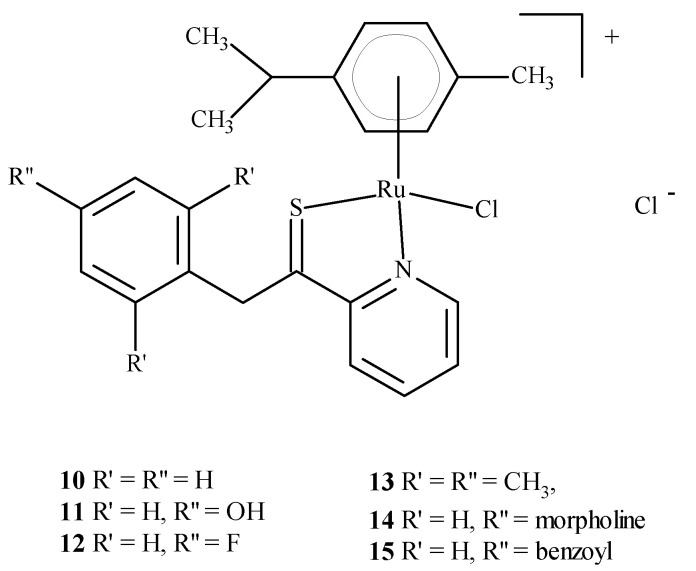
Structure of arene Ru(II) complexes with different substituents **10**–**15** [46].

## 3. Medicinal and Pharmaceutical Potential of Ru(II)/(III) Complexes

### 3.1. Application of Complexes of Ru(II)/(III) in Clinical Research

Drug combinations were found to demonstrate synergy of action, facilitate dose reduction and minimize drug resistance in clinical trials but also lead to adverse reactions [47]. Complex [Ru(KTZ)Cl_2_] (**16**) (where KTZ = ketoconazole) (Figure 5) showed a favorable synergistic effect in hormone-resistant cancer therapy [48]. This compound significantly increased the level of caspase-3 and promoted cell apoptosis in the metastatic melanoma cell line (C8161). The mechanism of action of **16** relies on its accumulation in the mitochondria and the activation of superoxide dismutase (SOD). Complex **16** inhibited cell growth in comparison to cisplatin (**1**) and showed the induction of apoptosis through the fragmentation of poly(ADP-ribose)polymerase (PARP) and stimulation of pro-apoptotic protein and Bcl-2 protein expression. Compound [Ru(KTZ)Cl_2_] (**16**) inhibited cell proliferation by the induction of epidermal growth factor receptor (EGFR) inhibitor, which resulted in the destruction of resistant nodular cells [49].

Berger et al. [51] investigated combinations of Ru(II)/(III) complexes and anti-cancer drugs. They showed that KP1339 (**19**) in combination with sorafenib was more efficient than sorafenib alone in the treatment of a human hepatoma cell line (Hep3B); sorafenib is an inhibitor of serine/threonine-specific protein kinase (Raf kinase); growth factor receptor beta derived from platelet (PDGFRβ); tyrosine–protein kinase (C-KIT); tyrosine kinase receptor of growth factors (RET); type 1, 2 and 3 of receptor vascular endothelial growth factor (VEGF) and Fms-related receptor tyrosine kinase3 (Flt-3).The combination of the two drugs was effective in cancer cells that were resistant to sorafenib. Intracellular accumulation of the drugs was observed, thus, influencing DNA replication, preventing cells from successfully entering mitosis and increasing apoptosis.

The administration of NAMI-A (**17**) (Figure 6) in combination with gemcitabine led to a reduced tolerance in comparison to gemcitabine alone and the inhibition of formation of non-small cell carcinoma cells. Anemia, renal dysfunction and neutropenia also occurred [52]. The drug combinations with synergistic potential were developed by Sava et al. through the use of high-throughput screening [53]. Furthermore, compound **17** (Figure 6) and doxorubicin showed potential synergistic efficacy. Complex **17** significantly increased doxorubicin accumulation in the breast cancer cells. Additionally, this combination of drugs led to 70% lower metastasis in mice with mammary carcinoma (MCa) in comparison with doxorubicin.

The promising synergistic effects of drugs were observed in a preclinical lung metastasis mouse model [53]. However, side effects were noticed when the maximum dose of the drug was used. The tumor vasculature is poorly organized, which affects the non-vascular penetration of drug molecules [54]. Drug absorption is negatively affected by oxygen supply and reduced blood flow, which prevents cancer treatment [55]. The combination of doxorubicin and RAPTA-C (**5**) (Figure 3) caused apoptosis of A2780 ovarian cancer cells compared to compound **5** alone [56].

The first Ru(III) compound to be clinically tested was complex **17** (Figure 6). Additionally, other ruthenium complexes, KP1019 (**18**), KP1339 (**19**) and TLD1433 (**21**) (Figure 6), are undergoing clinical trials [57,58,59]. NAMI-A (**17**) (Figure 6) entered phase II clinical trials, but its limited effectiveness resulted in the early closure of the trial. Compound **18** entered the first phase of clinical trials, but these were suspended due to poor solubility. However, studies of **19**, a sodium salt of **18,** continue. Compound TLD1433 (**21**) has passed the second stage of clinical trials on the non-invasive treatment of the urinary bladder with photodynamic therapy [8,57,58,60]. Another organometallic compound of Ru(II) showed potential anti-tumor properties [61]. Quite recently, Ru(II) complexes such as RAPTA-C (**5**) (Figure 3), RDC11 (**20**) (Figure 6) and RM175 (**22**) (Figure 6) passed phase I clinical trials [62].

Arene Ru(II) complexes, such as AH54 (**23**) and AH63 (**24**) (Figure 7), which are clinically approved, were also used as radiosensitizers in the Caco cells and **5** in A2780 cells [63]. The compounds were used as a pattern for the development of anti-tumor drugs because of their solubility, lipophilicity and stability [64].

RM175 (**22**), which is shown in Figure 6, contains an arene substituent, and that facilitates the penetration into cells and the binding to nitrogen at the N7 position of DNA. The arene substituent facilitates hydrophobic interactions of **22** (Figure 6) by intercalation with DNA. Moreover, **22** was shown to inhibit matrix metallopeptidase 2 (MMP2), which is important for metastasis due to the suppression of the immune system, the degradation of the extracellular matrix and the modulation of various pathways that facilitate tumor invasion and progression [65,66]. It also influences the neoplastic tumor microenvironment, growth and angiogenesis of neoplastic cells through various signaling pathways [66]. RM175 (**22**) (Figure 6) inhibited cellular translocation (haptotaxis) in the normal breast epithelial cell line (HBL100) and breast adenocarcinoma cells (MDA-MB-231). An alternative of RM175 (**22**) led to the formation of ONCO4417 (**25**) (Figure 8), which showed anti-tumor properties and cell cycle closure in the G2/M phase. The efficacy of this compound was similar to **1** (Figure 1) in various cell lines such as the ovaries, lungs, esophagus, pancreas, melanoma and colon. This compound showed strong activity against cisplatin resistant cells—A2780cisR.

Ruthplatin compounds **26**–**29** (Figure 9) are applied for treating tumor metastasis and drug resistance. The Pt(IV)-Ru(II) compounds are soluble in water. These complexes had anti-tumor activity in the 2D and 3D models of A2780cisR and A549cisR cells [67]. In addition, these compounds were cytotoxic towards breast cancer cells (MDA-MB-231, MCF-7), cervical cancer cells (HeLa) and promyelocytic leukemia cells (HL60) and are less toxic towards lung fibroblast cells (MRC-5) compared to **1**. These complexes were more active in comparison to the 3D model of MCF-7 nodular cells at lower concentrations of compounds (Figure 9) [67].

### 3.2. Antidiabetic Activity of Ru(II)/(III) Complexes

A metabolic disease such as diabetes type 2 is characterized by the decreased sensitivity of muscle tissues to fats and insulin, leading to poor blood sugar regulation [68,69]. This disease mostly results from diets high in carbohydrate and saturated fat and the associated decreased physical activity [68,70,71]. It is also known that the disease can be hastened by ROS. The structure of the Ru complex influences its substrate specificity. An example can be compound **30** (Figure 10), which competitively binds to glycogen synthase kinase (GSK-1β), an initiator of type 2 diabetes. The conversion of glucose to glycogen is catalyzed by multifunctional protein kinase, which has two isoforms, α and β, that differ in catalytic domains. The autophosphorylation of serine or threonine is an unfavorable effect of this enzyme. The similarity of arene complexes with staurosporine (**31**) (Figure 10) influences interactions with the ATP-binding cavity, where amino acids are located [68].

In addition, ruthenium complexes showed inhibitory activity for the protein dipeptidyl peptidase 4 (DPP-IV), which inhibits glucagon-like peptide-1 (GLP-1), which stimulates insulin release. The Ru(II) complex with p-cymene and ethylenediamine, [η^6^-p-cymene)Ru(en)]PF_6_^−^ (**32**) (Figure 11), showed interactions with sulfur donors [22]. The investigation on human protein tyrosine phosphatase (PTP-1B) found that it was less inhibited by complex **32,** possibly by forming a coordination bond with the sulfur atom at position 215 of cysteine.

### 3.3. Anti-HIV Properties of Ru(II)/(III) Complexes

Some research was carried out for the verification of anti-HIV activity of Ru(II) complexes. Ruthenium compounds are used for discovering new drugs to address problems connected with anti-HIV therapy. An analysis of the ability of Ru(II) complex with bipyridine (bpy) and eilatin-[Ru(bpy)_2_eilatin]^2+^ (**33**) (Figure 12) was performed. Eilatin is an alkaloid derived from the marine tunicate Eudiastoma sp. The anti-HIV activity against HeLa cells infected with HIV-1 was estimated with the use of a plaque-formation assay. The concentration needed to decrease HIV-1 activity by 50% during the mentioned assay was 0.8 µM and proves the significance of eilatin as a ligand [72].

The other complexes **34** and **35** (Figure 13) were analyzed as potential inhibitors of the HIV-1 and HIV-2 virus in human T Cell leukemia (MT-4). Complex **34** (Figure 13) showed an IC_50_ value higher than 0.21 µM, while the IC_50_ of complex **35** (Figure 13) was higher than 2.14 µM. However, none of them were selective and able to inhibit HIV viruses in MT-4 cells [73,74,75].

Carcelli et al. [75] synthesized quinolone derivatives, including structures **36** and **37**, shown in Figure 14, that partially retained the structural motif of elvitegravir, which is an inhibitor of HIV-1 integrase, as well as Ru(II) complexes, including [RuCl(η^6^-p-cymene)(**36**)] (**38**), [RuCl(η^6^-p-cymene)(**37**)] (**39**) and [RuCl(η^6^-hexamethylbenzene)(**37**)] (**40**) [73,75]. An assessment of the enzymatic activity of anti-HIV-1 integrase indicated that complexes (**38**) and (**39**) (Figure 15) have greater potential to inhibit HIV-1 integrase. The arene part of the structure related to Ru ions is not significant in the development of ligand activity.

Sheng et al. [76] discussed the interaction of the zinc finger of the nucleocapsid 7 (NCp7) with mononuclear compounds **17** (Figure 6), **41** and **42** (Figure 16) and the binuclear complexes **43** and **44** (Figure 16) of Ru(II)/(III). It is important in the HIV cell cycle to expose the influence of ligands on reactivity, as well as interaction with nucleic acids. Based on fluorescence spectroscopy and titration, the reactivity between Ru(II)/(III) complexes and NCp7 decreased according to the following sequence: **43** > **44** > **17** > **41** ~ **42** (Figure 16). The most reactive was compound **43**. The binding of Ru(II)/(III) ions to thiol groups of NCp7 is connected with an increase in the concentration of compound **43** and a decrease in its absorbance. The release of Zn(II) ions from NCp7 during interaction with complex **41** (Figure 16) depended on the concentration of **43** (Figure 16).

The removal of complex **43** (Figure 16) from NCp7, when Ru(III) binds to cysteine, terminates the protein folding process. An influence of the binding of Ru(III) ions on the interaction of spliced-leader 2 (SL2 DNA) with NCp7 indicates that the interaction of complex **43** (Figure 16) or complex **17** (Figure 6) with NCp7 inhibits recognition of protein by spliced-leader 2. Subsequently, it disrupts formation of the NCp7/SL2 complex by compound **43** (Figure 16) or **17** (Figure 6), which was found to be the most potent inhibitor in terms of concentration for compound **43** (Figure 16) [75].

### 3.4. Cytotoxicity of Ru(II)/(III) Complexes

In developing countries, cancer is the second most common fatal disease after heart issues. It is associated with an inappropriate lifestyle and a negative environmental effect on people’s health. Various treatments for cancer are currently available, such as chemotherapy, which blocks the activity of rapidly-proliferating cells. A lot of metal ion complexes have significantly influenced the development of anti-tumor drugs because of their use in diagnostics and therapy [1]. High-throughput screening indicated a strong potential for synergistic interactions between the EGFR inhibitor erlotinib and compound **5**. Moreover, no synergy was found in a previous review when sunitinib, sorafenib, BEZ-235 (Dactolisib) and an mTOR inhibitor (mammalian target of rapamycin), were tested at different dose ratios with RAPTA-C [77].

The cellular penetration analysis of ruthenium complexes showed that erlotinib significantly increased compound **5** uptake in the A2780 and A2780cisR cells compared to uptake of **5** alone. This investigation supports the finding that erlotinib and compound **5** interfere with the growth of different cell types; more apoptosis was observed in the umbilical vein endothelial cells (ECRF24) and human umbilical vein endothelial cells (HUVEC), but DNA cross connections preventing cytokinesis were observed in ovarian cancer cells. Complex **45** (Figure 17) was evaluated as a prodrug in photo activated chemotherapy (PACT), and compound **45** was found to be 1.2 to 1.4 times more toxic in HeLa cells compared to the RPE-1 cells, which are non-cancerous retinal pigment epithelial cells. The cytotoxicity of compound **45** increases after exposure to light. The relationship between the IC_50_ values in the dark and after exposure to light is defined as the phototoxicity index (PI). The value for compound **45** was 1.5–1.6 µM after one hour of incubation and 1.2 µM after four hours [78].

Ruthenium complexes with various bioactive ligands such as hormone receptors, growth factors and enzymes are commonly used as potential anticancer agents, especially against hormonal breast cancer and triple negative breast cancer (TNBC). Some of these complexes may be subject to multiple anti-tumor mechanisms. Cytotoxicity and anti-tumor properties can be induced in breast tumor cells by ruthenium complexes. Due to IC_50_ values less than 1 µM, arene Ru (II) complexes and Ru (II) complexes with cyclopentadienyl were proven to be of particular interest. Several ruthenium complexes have demonstrated low IC_50_ values in TNBC cells, which makes them attractive for further research [80]. The studies were carried out on the complexes of ruthenium with letrozole (**48**) (Figure 17), a non-steroidal aromatase inhibitor used in hormone therapy for breast cancer. Letrozole binds to Ru(II) ions through the nitrogen atom located in the triazole ring. Complex **49** (Figure 17) has effective anticancer properties towards MCF-7 cells but is less active against human glioblastoma cells (U251N). The cytotoxicity of complex **48** was significantly higher than that of the complex **46** (Figure 17), a RAPTA-C analog. The results of tumor cell exposure to **48** and 3-methyl adenine (**47**) or curcumin (present as a ligand in complex **49**) indicate the critical role of autophagy in inducing cell death by compound **49**. Cytotoxicity studies were performed using an MTT (3-(4,5-dimethylthiazol-2-yl)-2,5-diphenyltetrazolium bromide) test. After 24 h of incubation, the IC_50_ value for complex **48** against MCF-7 cells was 36 ± 6 µM [40]. After 72 h of incubation, the IC_50_ value for compound [Ru(p-cymene)curcumin)Cl]Cl (**49**) was 19.58 ± 2.37 µM. The IC_50_ value for complex [RuCl(Tp)(pta)(PPh_3)_] (**48**) where (Tp = trispyrazolylborate) was 11.3 ± 1.4 µM [81].

Complexes of Ru(II) ions with bidentate coumarins (**53**) and (**54**) (Figure 18) demonstrated greater cytotoxicity than ligands **50** and **51** against the tumor cell lines primary melanoma cells (WM115), HL-60 and -6 lymphoblastic leukemia cells (NALM) but were weaker than the reference compounds **1** and **2** [82].

**Figure 18 ijms-24-09512-f018:**
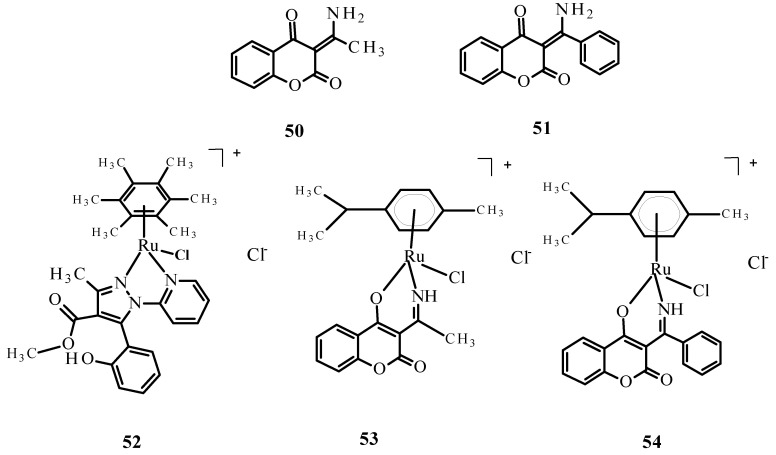
Structures of bidentate ligands **50** and **51** and their arene complexes witRu(II) **52**–**54**. Budzisz et al. [83] demonstrated that complex **52** also had cytotoxic activity, with IC_50_ values of 41.17 ± 3.68 µM for NALM-6 and 59.3 ± 4.8 µM for human colon adenocarcinoma (COLO-205). This activity was better than that of quercetin (IC_50_ 77.1 ± 7.8 µM against NALM-6). In addition, the comet assay [83] indicated that complex **52** causes DNA damage in NALM-6 (Figure 19).

**Figure 19 ijms-24-09512-f019:**
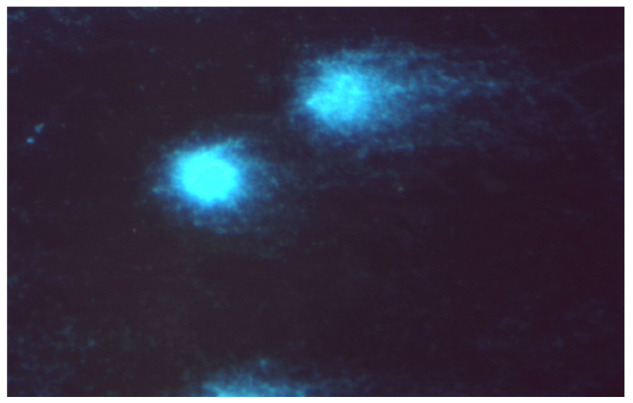
Damaged DNA of comet-shaped NALM-6 cells after treatment with complex **52** at a concentration of 50 µM (stained with DAPI, magnification 400×).

### 3.5. Ru(II)/(III) Complexes with Potential Anti Alzheimer’s Disease Properties

Alzheimer’s disease has a complex, multifactorial and neurodegenerative pathogenesis. Other pathologies of the central nervous system, such as bovine spongiform encephalopathy (BSE) or Creutzfeldt–Jakob disease (CJD), are referred to as prion diseases, which are associated with alterations in the metabolism of the neurotoxic and amyloidogenic prion peptide (PrP 106–126) and its accumulation [84]. None of the drugs currently used in medicine alter the inevitable progression of Alzheimer’s disease. With respect to prion peptide PrP, complexes **17** and **18** are able to bind to the sequence of PrP 106-126 and inhibit its aggregation; however, compound **18** is more effective because it has a larger aromatic ring in its structure. In addition, compounds **17** and **18** influenced the oxidation-reduction potential of Cu(II) ions and inhibited ROS generation [84]. The complex of Ru(II) with polypyridine ligand **55** (Figure 20) can reduce the toxic effect of β-amyloid, because complex **55** is able to trap Cu(II) ions, which cause the formation of β-amyloid aggregates. The findings showed that complexes of Ru(II) mostly inhibit the accumulation of β-amyloid aggregates, scavenge intracellular ROS, reduce mitochondrial degradation and maintain the correct level of matrix metalloproteinases (MMPs). These results strongly testify to the correct design of these complexes in terms of their mechanisms of action. The Ru(II) complex with polypyridine ligand **55** (Figure 20) may reduce the toxic effects of β-amyloid because it has the ability to trap Cu(II) ions, which cause the formation of β-amyloid aggregates. The experimental analyses performed on neuroblastoma tumor cells (SH-SY5Y) demonstrated that compound **55** significantly inhibits amyloid β aggregation, scavenges ROS, limits mitochondrial degradation and maintains the correct mitochondrial membrane potential (MMP). These results largely prove the correct design of the structure of this complex in terms of the mechanism of action [85].

The density functional theory (DFT) calculations and in vitro experiments confirmed that the arene Ru(II) complex with curcumin **49** (Figure 17) is capable of inhibiting the accumulation of two β-amyloid molecules: one peptide composed of 1–40 amino acids and another composed of 1–42 amino acids. The final results indicate that complex **49** moderately affects the aggregation of β-amyloid through the hindering effect [86].

### 3.6. Photosensitizing Complexes of Ru(II)/(III)

Compounds of ruthenium, especially those with an oxidation state (II), are potential photosensitizers in PDT therapy; they possess physicochemical and electrochemical activities, stable reaction kinetics, low toxicity and specific intercellular transport through transferring chemical groups by specific catalytic enzymes. The complex of Ru(II) with 2,2-bipyridine, **56**, is a reference compound for the evaluation of Ru(II) compounds with photophysical and photochemical activity. Excitation takes place at around 420 nm and the molecule enters a triplet excitation state (^3^MLCT) in oxidized acetonitrile with a lifetime of 200 ns and 76 µs in unoxidized acetonitrile. However, the fluorescence quantum yield, with ^1^O_2_ formation, is 10% in unoxidized acetonitrile and 56% in oxidized acetonitrile.

A complex of Ru(II) with dipyrido [3,2-a:20.30c]phenazine and 2,2-bipyridine (**57**) is unique as the ligand positively influences cellular uptake. This complex exhibited effective phototoxicity at 625 nm and luminescent activities typical of the triplet excitation state. The exposure of metal ion compounds to UV light causes an exchange of ligands and photoactivation and leads to the formation of covalent bonds with DNA. Hence, compound **57** could be a luminescent marker of DNA. It could be applied in photodynamic therapy as a photosensitizer due to ROS formation. A strong phototoxic effect is achieved by prolonging the triplet excitation state. Recently, an investigation of differences between compounds **57** and **58** (Figure 21) (which contain benzo[i]dipyrido [3,2-a:20,30-c]phenazine (dppn)) was conducted. It was found that the length of the excitation state for compound **57** in acetonitrile was approximately 33 µs. The extended triplet excitation state results in a strong phototoxic effect. It was also demonstrated that the presence of ligands such as 4,7-diphenyl-1,10-phenanthroline (dip); 1,10-phenanthroline (phen); 2,3-bis(2-pyridyl)pyrazine (bpp) and 2,2-bipyridine significantly influence photoactivity. Pure compound **57** demonstrated a phototoxicity index (PI) value above 150 and its analog with methoxy substituents yielded a PI of approximately 43. The derivatives caused strong breaks in plasmid DNA when exposed to radiation. Pyridine-porphyrin systems substituted in three places show stronger phototoxic properties than systems substituted in four places [87].

To evaluate the influence of aromatic elements on anti-tumor activity, Therrien et al. [87,88] modified Ru(II) complexes by adding a meso-4′-tetrapyridine-porphyrin substituent. At a concentration of 10 µM, the compounds demonstrated 60–80% cytotoxicity against human melanoma cells (Me300), measured at 652 nm. Complexes **59** and **60** (Figure 22) exhibited the strongest phototoxicity in HeLa cells irradiated with 8 mW at 850 nm. After irradiation, the compound **60** changed its localization in the cell, resulting in membrane destruction and cell death [87].

The design of selective photosensitizers for anti-tumor therapy is still a challenge. The intra-system transition of electrons increases due to spin-orbital coupling. A low fluorescence quantum yield indicates the possibility of singlet oxygen generation and intra system transition in these molecules. Photoactivation methods are used in anti-tumor therapy [10]. Photosensitizing molecules are administered intravenously or topically in the treatment of PDT [10,89,90,91,92]. After the introduction of a drug into neoplastic cells, the tissue target is irradiated at a specific wavelength of light [10,93]. An excited compound activates oxygen or biomolecules and generates ^1^O_2_, which contributes to the death of cancer cells. Two-photon excitation (TPE) and a longer wavelength are usually chosen for PDT in case the light has to reach deeper layers [10,94,95]. Liu et al. [10,35] described Ru(II) polypyridine complexes, which caused the formation of singlet oxygen in methanol, mitochondrial accumulation and two-photon absorption (TPA). These compounds activated HeLa cell death by releasing singlet oxygen after one or two irradiations but were not toxic in the dark. Complex **61** (Figure 23) is the most promising of these, because the IC_50_ value towards 3D multicellular spheroids was 9.6 µM in the case of single-photon PDT and 1.9 µM for two-photon PDT therapy. Both values were lower than for compound **1**.

Chen et al. [96] developed arene complexes of Ru(II) as drugs for PACT and PDT therapies. Compound [(p-cymene)Ru(dpb)(py)]^2+^ (**62**) (Figure 24) actively absorbed long-wavelength light and generated singlet oxygen, causing DNA photo-cleavage. In addition, complex **62** breaks down into pyridine (py) and 2,3-bis(2-pyridyl) benzoquinoxaline (dpb) upon irradiation. The remaining part, Ru-(p-cymene), binds to DNA bases. For this reason, complex **62** may be considered a potential anti-tumor agent used in PDT and PACT methods. The dissociated dpb ligand fluoresces and allows real-time monitoring of interactions between the photoactivated complex and biomolecules. Significant cytotoxicity after irradiation towards A549 cells showed complex **62** with an IC_50_ of 4 µM. In the dark, the IC_50_ value was 27.6 µM.

Moreover, Huang et al. [10,97] described complexes of Ru(II) with terpyridine (Figure 25) as photosensitizers in PDT therapy and studied their phototoxicity. These complexes were localized in the nucleus; they luminesced between 670 and 710 nm and generated singlet oxygen and radicals. The most active complex, **63** (Figure 25), showed significant phototoxicity with a low IC_50_ value towards human liver cancer cells (Bel7402) and human liver cancer cells (HepG2) compared to **1**. Frei et al. [10,98] described polypyridyl complexes such as Ru(dip)_2_(bdt) (**64**) (where dip= 4,7-diphenyl-1,10-phenanthroline; bdt=1,2-Benzenedithiolate) and [Ru(dqpCO_2_Me)(ptpy)]^2+^ (**65**) (where dqpCO_2_Me, 4-Methylcarboxy-2,6-Di(Quinolin-8-yl)Pyridine; ptpy, 4′-Phenyl-2,2′:6′,2″-Terpyridine) (Figure 25). Complex **64** had a low IC_50_ value measured for phototoxicity after the irradiation of HeLa cells [10,99]. Approximately 67% of **65** accumulated in the mitochondria. Both compounds were tested microbiologically against strains such as *Staphylococcus aureus* and *Escherichia coli*. Complex **64** (Figure 25) was effective in reducing the survival rate of Gram(+) cells, Staphylococcus aureus, but no activity was observed against *Escherichia coli*. Compound **65** (Figure 25) was effective in reducing cell survival against *Staphylococcus aureus* and *Escherichia coli*.

Karaoun et al. [10,100] prepared Ru(II) complexes with econazole for use as a cell marker and factor in PACT. The complexes showed luminescence in the dark and were stable. Econazole (Ec) release was induced by irradiation with green light and terminated the luminescence. Complex [Ru(phen)_2_(Ec)_2_Cl_2_]^2+^ (**66**) (where phen = phenanthroline) (Figure 26), showed a higher PI value than the analogical compound with a single econazole ligand. Complex [Ru(dppz)_2_(CppH)]^2+^ (**67**) (where dppz = dipyrido[3,2-a:2′,3′-c]phenazine; CppH = 2-(2′-Pyridyl)pyrimidine-4-carboxylic Acid) (Figure 27) [10,101] disturbed the proper functioning of the mitochondria. Complex **67** is formed from complex [Ru(dppz)_2_(CppDMNPB)]^2+^ (**68**) (where DMNPB = 3-(4,5-Dimethoxy-2-nitrophenyl)-2-butyl esters)) (Figure 27) after 24 h in the shade, and 7% of this complex was hydrolyzed in PBS buffer.

### 3.7. Computational Approaches to Studying Interactions of Ru(II)/(III) Complexes with Their Biological Targets

Experimental drug design is time consuming and very costly. It is, thus, not surprising that computational methodologies are permeating all aspects of the drug discovery process. The best approach would be to carry out quantum-mechanical (QM) calculations on interactions between a molecular target (e.g., enzyme) and a ligand (e.g., inhibitor). At present, however, this approach has limited application in drug design because of the length of time and huge computational resources needed. For this reason, the most popular method of studying receptor–ligand interactions remains rooted in molecular mechanics (MM) based on classical mechanics and molecular docking [102,103,104,105]. The use of only non-bonding components of MM energy allows for fast evaluation of the interaction energy as a function of different geometries. The geometry of the receptor is usually held rigid, and only the conformation of the ligand is varied in the search for the best fit to the receptor. This is followed by evaluating the binding affinities of these conformations (called poses) to the receptor. The enormous speed of molecular docking allows huge libraries of small molecules to be searched at low cost to identify potential drug candidates; the procedure, called virtual screening (VS), has become an elementary step in modern drug design [106,107,108,109,110,111,112,113].

Over the years, interim computational schemes emerged, in which docking results are further scrutinized at higher theory levels. The most popular approach treats the bulk part of the receptor at the MM level, while the ligand and its neighboring parts of the receptor are treated at the QM level (QM/MM methodology). The remainder of this chapter will discuss the move from molecular docking techniques, to QM/MM, to QM levels of theory for investigating ruthenium complex interactions.

Numerous attempts were made to predict and/or rationalize (in terms of types of intermolecular interactions) ruthenium complexes interactions using molecular docking. It should be kept in mind, however, that the theoretical tools behind docking are simplified so the results usually vary depending on the particular function describing the energy of binding (called score) and should be used cautiously. Furthermore, molecular docking relies heavily on the parameters. These are sometimes not available for Ru or made ad hoc without sufficient testing. As a result, the recent studies use a range of docking programs, apparently based more on their availability rather than benchmark performance. For example, six different docking programs (Autodock, Vina, Cdocker, Sybyl, Dock and Hex) were used in seven recently published articles [79,114,115,116,117,118,119]. Another problem arises from the fact that docking to DNA [120,121] is fairly new and unexplored. One way around the lack of reliable parameters is to use iron complexes in place of those that contain ruthenium. This procedure was validated by showing that the binding modes of both complexes are identical (see Figure S3 in reference [122]). In these studies, it was shown that the results obtained with the Gold program are in better agreement with the experiment than those obtained with Vina, and that the ruthenium complexation in the DNA minor groove is energetically favored over intercalation, as was also observed for ferrocene appended Ru(II), Rh(III) and Ir(III) dipyrrinato complexes [123].

The methodological aspects of docking are nicely illustrated by studies comparing three very popular docking protocols; Adeniyi and Ajiba [124] performed a benchmark of three computational tools implemented in Autodock, Gold and Glide to predict the applicability of different ruthenium complexes in cancer therapy. They examined 21 ruthenium complexes belonging to the RAPTA family and docked them in eight different proteins: BRAF kinase, cathepsin B (CatB), histone deacetylase (HDAC7), histone protein in nucleosome core particle (NCP), ribonucleotide reductases (RNR), recombinant human albumin (rHA), thioredoxin reductase (TrxR), thymidylate synthase (TS) and topoisomerase II (Top II). In all three programs, CatB was the most promising target for these complexes, whereas TrxR, TS and rHA were predicted to be average or rare. All three programs also showed that the RAPTA complex binding was enhanced when hydrolyzed. These results agree qualitatively with the experimental findings. Results from Glide were worse in this regard than those from the other two programs; this was attributed to the overestimation of the steric hindrance by the scoring function of Glide.

A ruthenium(II) complex **69** (EPIRUNO_2_) (Figure 28) was also studied as a potential drug for the parasitic disease leishmaniasis [125]. Araújo et al. [126] performed molecular docking of a complex with imidazole alkaloid abundant from the leaves of Pilocarpus microphyllus, epiisopiloturine (EPI) and nitrogen(IV) oxide (**69**) in nucleoside diphosphate kinase (NDK) from Leishmania amazonensis. The results were compared against a known drug, miltefosine.

They reported binding energies –31.76 and –11.30 kJ/mol and inhibition constants of 2.70 and 10.57 μM for (**69**) and miltefosine, respectively. In another study [127], they docked compound (**69**) in five different targets of Leishmania major: glycyl peptide N-tetradecanoyltransferase (PDB code: 5g20), UDP-glucose pyrophosphorylase (PDB code: 5nzg), nucleoside diphosphate kinase (PDB code: 5c7p), pteridine reductase (PDB code: 1e7w) and nucleoside hydrolase (PDB code: 1ezr). The highest affinity was observed for 1e7w (binding energy –44.69 kJ/mol and inhibition constant 14.80 μM) with four hydrogen bonds being formed and for 5nzg (binding energy –43.97 kJ/mol and inhibition constant 19.74 μM) with three hydrogen bonds. The modes of these bindings are illustrated in Figure 29. The 1e7w protein is essential for the growth of the parasite, and 5nzg is vital for the production of cell surface glycans and other processes that are responsible for the pathogenicity of a parasite, thus, making (**69**) a very potent drug candidate to combat the spread of leishmaniasis.

Das and Mondal [128] docked three ruthenium(II) complexes: Ru(tmp)_2_(dpq)]^2+^ (**70**), [Ru(tmp)_2_(dppz)]^2+^ (**71**) and [Ru(tmp)_2_(11,12-dmdppz)]^2+^ (**72**), where dppz = dipyrido[3,2-a:20,30-c]phenazine, dmdppz = dimethyldipyrido[3,2-a:20,30-c]phenazine, dpq = dipyrido[3,2-d:2′3′-f]quinoxaline, and tmp = 3,4,7,8-tetramethyl-1,10-phenanthroline (Figure 30), to two B-DNA hexamers of alternative AT and GC sequences: d(ATATAT)_2_ and d(GCGCGC)_2_.

The results showed that the complexes intercalated into the minor groove of DNA through the diamine ligand and preferred to bind to the d(ATATAT)_2_ sequence. Complex (**72**) exhibited higher affinity than complexes **70** and **71**. The QM/MM approach provided information about the stability of the adducts, showing the highest stability in the case of the complex (**72**) and the d(ATATAT)_2_ sequence. The binding mode of **70** is illustrated in the Figure 31. The results also suggested that a high number of intercalating aromatic rings increased the binding affinity towards the DNA receptors.

As it was already mentioned, molecular docking studies are often supplementary to the experimental results. Paitandi et al. [117] synthesized four new arene ruthenium(II) complexes: [(η^6^-C_10_H_14_)RuCl(MFPdpm)] (**73**), [(η^6^-C_12_H_18_)RuCl(MFPdpm)] (**74**), [(η^6^-C_10_H_14_)RuCl(PFPdpm)] (**75**) and [(η^6^-C_12_H_18_)RuCl(PFPdpm)] (**76**), where MFPdpm = 5-(4-fluoro)phenyldipyrromethene, and PFPdpm = 5-(penta-fluoro)phenyldipyrromethene (Figure 32). All four compounds exhibited significant toxicity towards the A549 cell line with the best performance of (**76**). Molecular docking revealed that fluorine atoms of (**76**) were engaged in hydrogen bonding interactions with DNA nucleobases, and the estimated binding energy was the lowest for (**76**). Additional density functional theory (DFT) calculations used to estimate the LUMO energies of the complexes also indicated that (**76**) was the most reactive towards the DNA.

In similar studies Chen et al. [129] investigated the binding of two ruthenium polypyridyl complexes [Ru(bpy)_2_(ptpn)]^2+^ (**77**) and [Ru(phen)_2_(ptpn)]^2+^ (**78**), where bpy = 2,2-bipyridine, ptpn = 3-(1,10-phenanthroline-2-yl)-as-triazino[5,6-f]1,10-phenanthro line, and phen = 1,10-phenanthroline (Figure 33) to human telomeric d[(TTAGGG)_n_](HTG22) quadruplex as a potential stabilizing agent. They used d(AG_3_[T_2_AG_3_]_3_) to model the DNA sequence with K^+^ and Na^+^ ions as receptors. Studies revealed that both complexes shared similar binding sites, and binding energy (obtained using molecular mechanics/generalized Born surface area, MM/GBSA method [130]) showed that the intercalation of [Ru(bpy)_2_(ptpn)]^2+^ in the presence of potassium (−234.4 kJ/mol) or sodium (−269.9 kJ/mol) cation exhibits lower binding energy than external stacking (−163.6 kJ/mol in case of potassium) or nonspecific binding (−118.8 kJ/mol in case of sodium).

A number of studies included subsequent molecular docking DFT calculations to further investigate the interactions of ruthenium complex–drug target systems and to obtain reliable estimates of binding energies and binding sites. Examples of such a protocol include half-sandwich ruthenium(II) complexes with human serum albumin (HSA) [131]; nitrosyl ruthenium(II) complex with bovine serum albumin (BSA) [132]; Ru(II)-arene complexes with DNA and BSA [115]; ruthenium(II) polypyridyl complexes with β-amyloid peptide [19] and DNA [133,134] cyclopentadienyl-; cyclooctaniedyl-based ruthenium(II) and anastrozole-, and letrozole-based ruthenium(III) complexes with human aromatase [135]; Ru(III) chelated with (E)-2-((phenylamino)-N-(pyridine-2-yl)methylene)acetohydrazide ligand bound to lanesterol 14 alpha-demetyhlase (CYP51) enzyme of *E. coli* as a candidate for an antibacterial and antioxidative agent [136], a dinuclear Ru(II) complex; [(Ru(phen)_2_)_2_-(tpphz)]^4+^ (where phen = 1,10-phenanthroline, tpphz = tetrapyridophenazine) with G-actin monomers to block its transformation to F-actin filaments [137]; ruthenium(II) containing phenazine (or phenanthroline) rings intercalating the DNA (and also as DNA “light switches” [138]) and bound to DNA topoisomerase I [139,140]; novel Ru(II)-arene complexes with non-steroidal anti-inflammatory drugs bounded to the COX-2 enzyme [141,142], as well as HSA and DNA [143]; photoreactive Ru(II) complexes with telomeric G-Quadruplex DNA [144]; Ru(II)-Schiff base complexes inhibiting HepG2 cells [145]; ruthenium(II)-bipyridine-calixarene complex with BSA and ovalbumin [146]; and ruthenium(II)-purine complexes with BSA [122].

In the overwhelming majority, the molecular docking of ruthenium complexes is performed as an auxiliary study, complementing and confirming the experimental findings. There are, however, theoretically-focused studies that mostly describe the interactions of ruthenium complexes with proteins. Patel et al. [147] performed an investigation with ruthenium complexes to inhibit *P. falciparum* calcium-dependent protein kinase 2 (PfCDPK2), which was shown to be crucial in the development of male gametes [148]. They examined two Ru(II) complexes, octahedral Ru-pyridocarbazole (FLL) (**79**) and methylated Ru-pyridocarbazole (E52) (**80**) (Figure 34), utilizing a combined ensemble docking (an approach in which an “ensemble” of the drug target is created to take into account its flexibility; this allows the performance of a docking with different initial conformations of the drug target) and molecular dynamics study.

The results obtained for both complexes were compared with those of a known inhibitor of PfCDPK2, staurosporine (STU) (**31**). The binding energies were estimated using the molecular mechanics/Poisson–Boltzmann surface area method [130]. They reported binding energies for STU, FLL and E52: –66.5, –67.2 and –91.3 kJ/mol, respectively, indicating that both FLL and E52 are a promising drug candidates to limit the spread of malaria.

Liu et al. [149] performed molecular docking of two enantiomeric analogs of E52 (**81a**) and (**81b**) (Figure 35) to three different protein kinases: GSK3-β, CDK2/cyclin A and PIM1, as a potential cancer-oriented drug.

They reported that the racemic mixture of **81a** and **81b** binds in a similar fashion to PIM1 and GSK3-β, but in the case of CDK2/cyclin A, they observed a novel conformation with a marked difference in binding. This was explained by the different topology of the active site at the location of PHE80 in CDK2/cyclin A; this stabilized the ruthenium complexes by a possible stacking interaction with the cyclopentadiene (Cp) ring and the lack of a hydrogen bond between the inhibitor and the linker region residue of the enzyme.

Das et al. [150] conducted QM/MM calculations and the docking of ruthenium(III) complexes in HSA. They considered four different monoaqua and diaqua complexes of NAMI-A: [trans-RuCl_3_(H_2_O)-(3H-imidazole)(dmso)] (**82**), [trans-RuCl_2_(H_2_O)_2_(3H-imidazole)(dmso)]^+^ (**83**), [trans-RuCl_3_(H_2_O)(4-amino-1,2,4-triazole)(dmso)] (**84**) and trans-RuCl_2_(H_2_O)_2_(4-amino-1,2,4-triazole)(dmso)]^+^ (**85**), where dmso = dimethyl sulfoxide (Figure 36). The results indicated that diaqua adducts, (**83**)-HSA and (**85**)-HSA, exhibited a higher binding affinity than the corresponding monoaqua adducts, (**82**)-HSA and (**84**)-HSA. The following residues of the active site play a crucial role in binding with the ruthenium complexes: Ala194, Arg145, Arg197, Asp108, Gln459, Glu425, His146, Lys190, Phe149, Pro147 and Tyr148. The QM/MM calculations revealed that **83**-HSA is the most stable adduct; however, (**85**)-HSA exhibits higher reactivity towards the protein.

We conclude this short review of the computational techniques used to study the interactions of Ru(II)/Ru(III) complexes with biological targets. It includes an example of QM calculations, which, as pointed out at the beginning, are much more demanding but, at the same time, much more reliable; however, such attempts are becoming more common nowadays due to the increasing power and accessibility of high-performance computers.

Extensive theoretical studies on nine different Ru(II) complexes of type mer-[Ru(Cl-Ph-tpy)(N–N)(X)]^+^ (X = F, Cl, Br; N–N = 1,2-diaminoethane, 1,2-diaminocyclohexane, or 2,2-bipyridine) (Figure 37) were carried out by Erkan et al. [151]. They evaluated different DFT levels of theory to assess the spectral properties of the complexes, investigate their electronic properties and predict their behavior in relation to the drug target. The molecular docking into the HeLa human cervix carcinoma cell line revealed the potential application of the complexes as anticancer drugs. The results revealed that all nine complexes interact well with the protein via five binding modes: hydrogen bonds, polar and hydrophobic interactions, pi–pi stacking, and halogen bonds. It should be mentioned that only three of the complexes tested (namely, **87**, **90** and **93**) had been obtained synthetically before; therefore, the study allowed us to verify if analogs of the existing drugs also exhibit anticancer effects, showing that all nine shared very similar binding energy values. All the non-synthetical complexes exhibited inhibition constants higher than 1 μM (whereas they were 816.0, 522.5 and 742.7 nM for **87**, **90** and **93**, respectively), apart from **89,** for which the value was estimated to be 645.0 nM.

## 4. Conclusions

This review article examines the potential medical and pharmaceutical applications of ruthenium complexes as antidiabetic, anti-HIV, anti-Alzheimer’s and anti-cancer agents, providing examples of specific complex compounds and their mechanism of action. In addition, examples of the Ru(II)/(III) complex compounds for which clinical trials were undertaken are also given. Ru(II)/(III) complexes are very important in research on coordination compounds with therapeutic activity, because they are an alternative to Pt(II) complexes. This is essential because of the fact that Pt(II) complexes, despite their use in the treatment of neoplasms, have many side effects. Hence, ruthenium has become a special point of interest in the development of metallopharmaceuticals. It should also be mentioned that Ru(II)/(III) is characterized by many interesting mechanisms of action, e.g., activation through reduction. It can also mimic Fe(III) ions and bind to transferrin, which are typical for Ru(III) complexes. Furthermore, ruthenium complexes demonstrate epigenetic activities, e.g., interaction with biomolecules, including coordination to the nucleosome core, formation of adducts with histones and inhibition of topoisomerase II. This results in the disruption of DNA replication, penetrating deep into the endoplasmic reticulum and accumulating in the mitochondria, which leads to oxidative stress and apoptosis of pathological cells. For this reason, many syntheses and their modifications are performed, taking into account the therapeutic goals, and hence, the structure of biomolecules.

A challenge for scientific researchers is the possibility of using Ru(II)/(III) complex compounds as a marker in cancer phototherapy. This method brings great promise due to the fact that ROS ultimately destroys cancerous cells. Some of the complexes contain ligands that are commonly found in plants, such as coumarins, flavonoids, chromones, indoles and alkaloids. So far, many derivatives of these compounds have been synthesized. In many cases, the biological activities of this type of ligands, used for medical and pharmaceutical reasons, are enhanced by complexation with Ru(II)/(III) ions.

The structure of other complexes of Ru(II)/(III) is similar to that of Pt(II) anticancer compounds and often includes an arene substituent that provides a piano-stool type of geometry, which increases the chance of interacting with the DNA of cancer cells. The complexes used as phototoxic agents in chemotherapy contain polypyridine ligands or nitrogen-substituted polycyclic aromatic hydrocarbons.

The computational methodologies, particularly molecular docking, in the study of Ru(II)/Ru/(III) complexes (or drug candidates in general) presented herein merely serve to illustrate the progress made in this approach and are by no means exhaustive. They show the importance of theoretical chemistry in modern drug design, whether virtual screening is used to prescreen a huge library of potential drug candidates and drug targets, to further investigate the interactions between the drug candidate and target or just to support experimental findings.

Despite numerous scientific projects, the main aim is to receive Ru(II)/(III) complexes, which, due to the modifications and properly selected ligands, increases the chances of their real use for medical and pharmaceutical purposes.

## Figures and Tables

**Figure 2 ijms-24-09512-f002:**
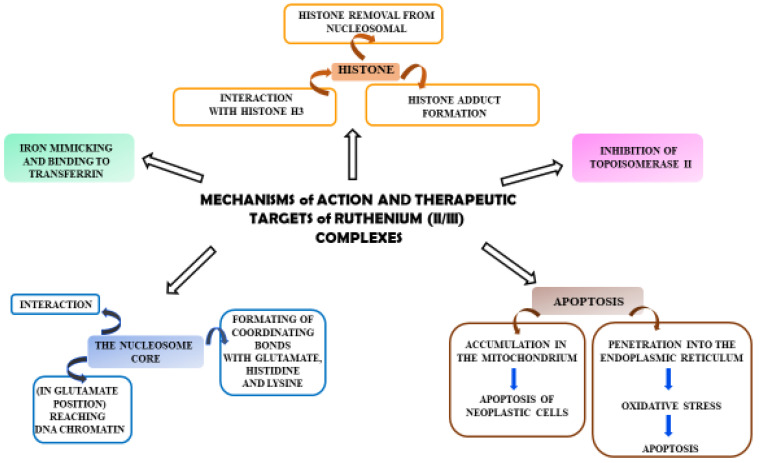
Diagram showing the mechanisms of action of Ru(II)/(III) complexes and their possible therapeutic targets in anti-tumor activity.

**Figure 3 ijms-24-09512-f003:**
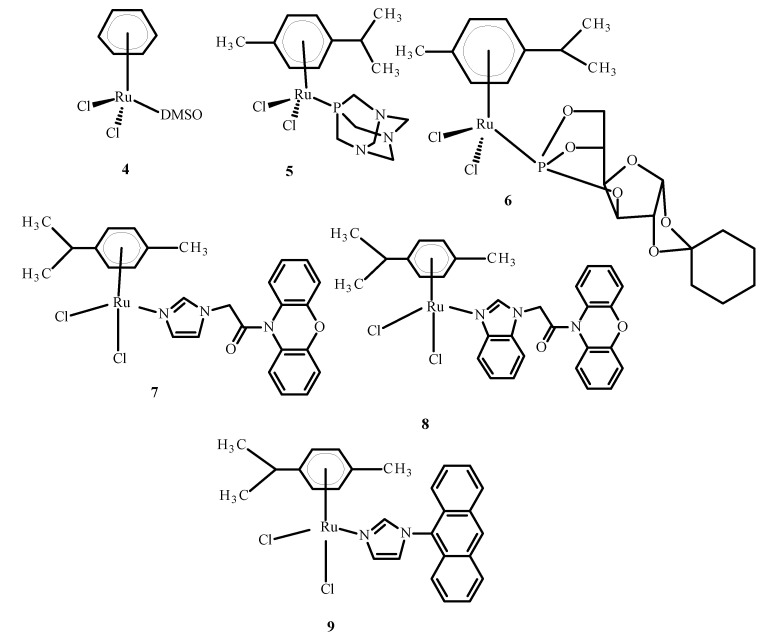
Structures of arene Ru(II) complexes [21].

**Figure 5 ijms-24-09512-f005:**
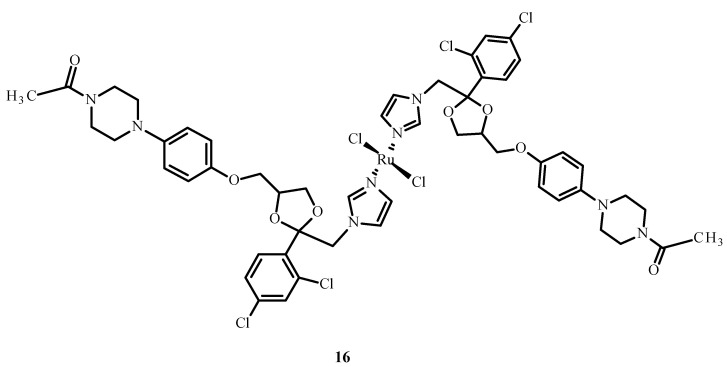
Structure of a complex [Ru(KTZ)Cl_2_] (**16**) [50].

**Figure 6 ijms-24-09512-f006:**
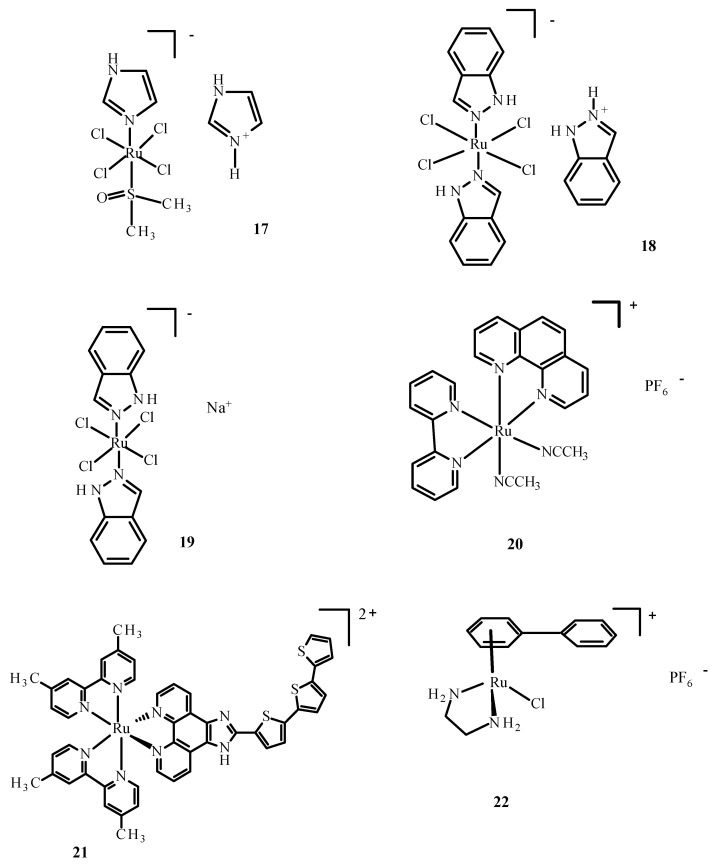
Structures of NAMI-A (**17**); KP1019 (**18**); KP1339/IT139 (**19**); RDC11 (**20**); TLD1433 (**21**); RM175 (**22**) [10].

**Figure 7 ijms-24-09512-f007:**
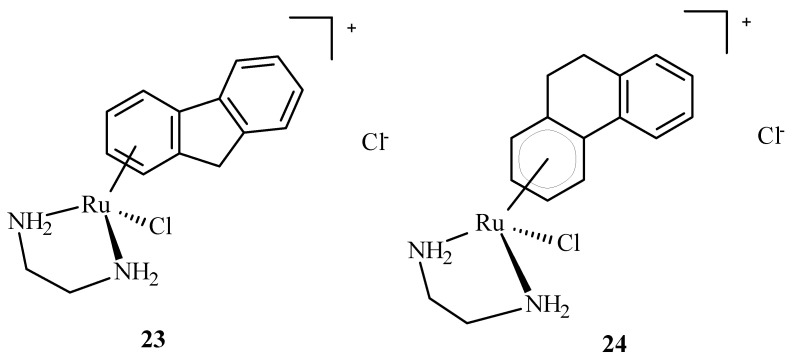
Structures of arene Ru(II) complex compounds AH54 (**23**) and AH63 (**24**).

**Figure 8 ijms-24-09512-f008:**
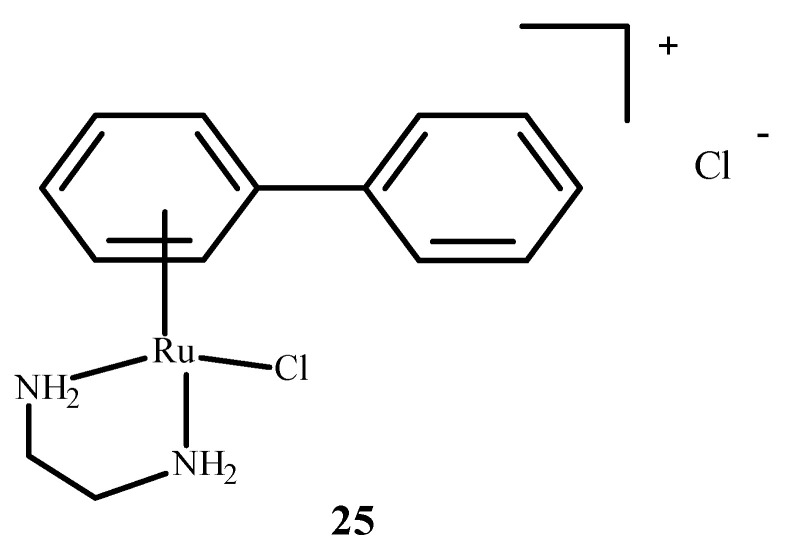
Structure of arene Ru(II) complex—ONCO4417 (**25**) [1].

**Figure 9 ijms-24-09512-f009:**
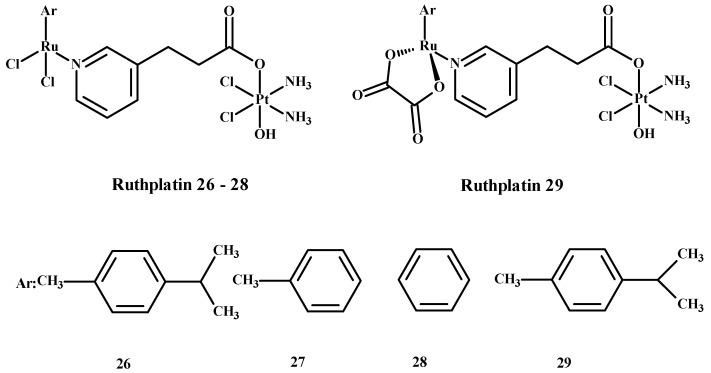
Ruthplatin complexes **26**–**29**. Ar—arene [1].

**Figure 10 ijms-24-09512-f010:**
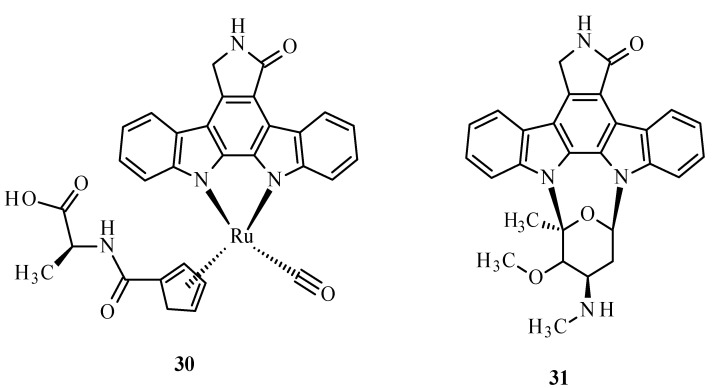
Structure of Ru(II) complex with modified staurosporine (**30**) and staurosporine (**31**) [68].

**Figure 11 ijms-24-09512-f011:**
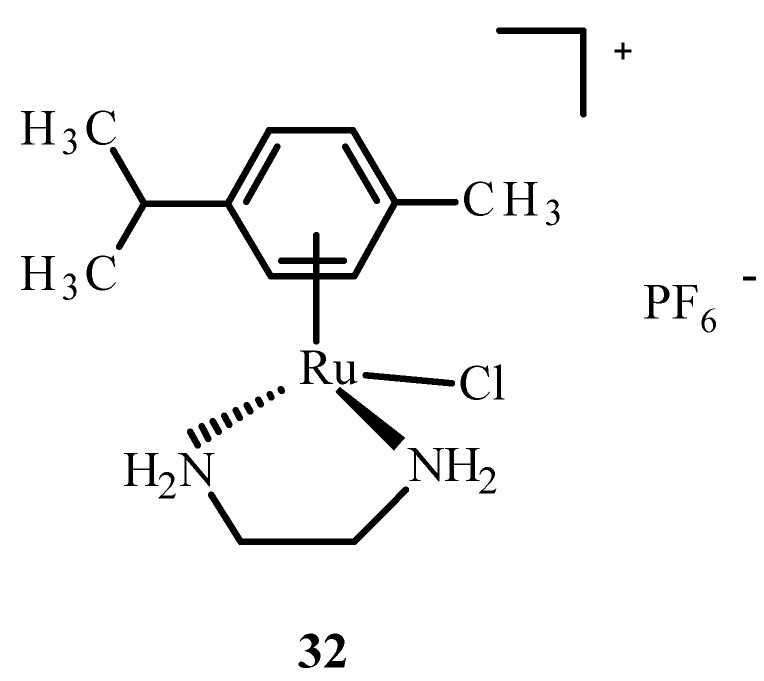
Structure of [(η^6^-p-cymene)Ru(en)Cl]PF_6_^−^ (**32**); (en-ethylenediamine).

**Figure 12 ijms-24-09512-f012:**
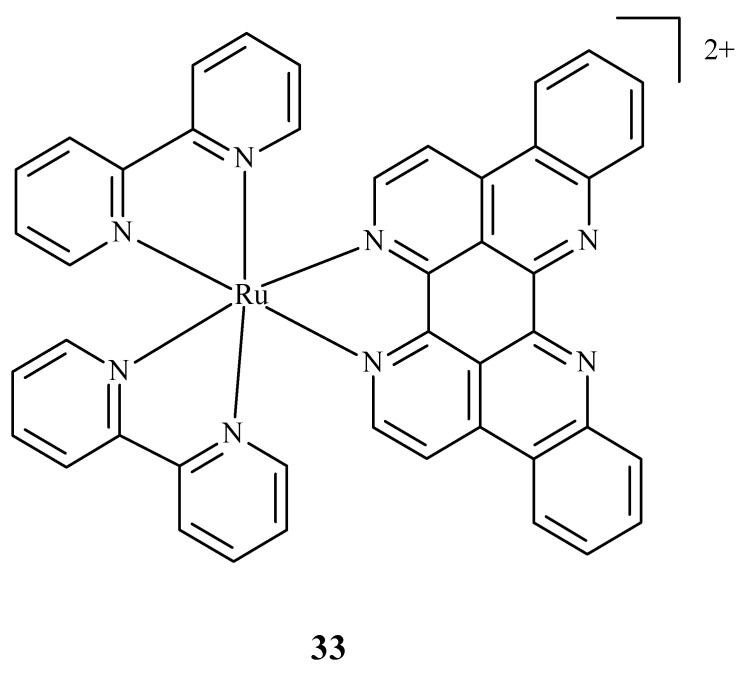
Structure of Ru(II) complex with bipyridine and eilatin **33** [72].

**Figure 13 ijms-24-09512-f013:**
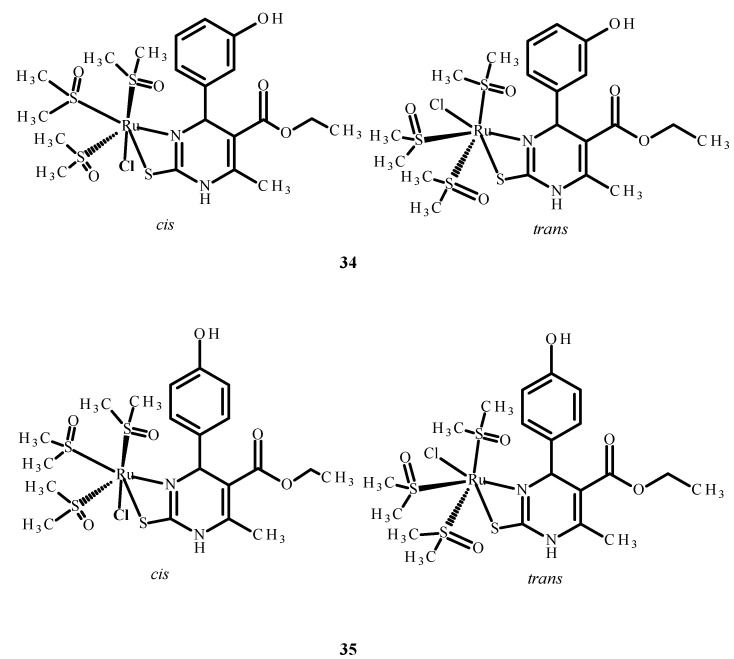
Structures of Ru(II) complexes **34** and **35** as cis and trans isomers [73].

**Figure 14 ijms-24-09512-f014:**
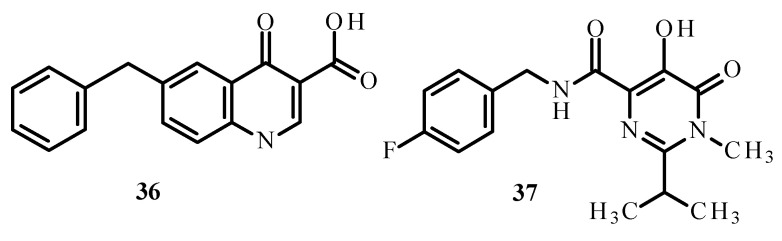
Structures of ligands **36** and **37** [73].

**Figure 15 ijms-24-09512-f015:**
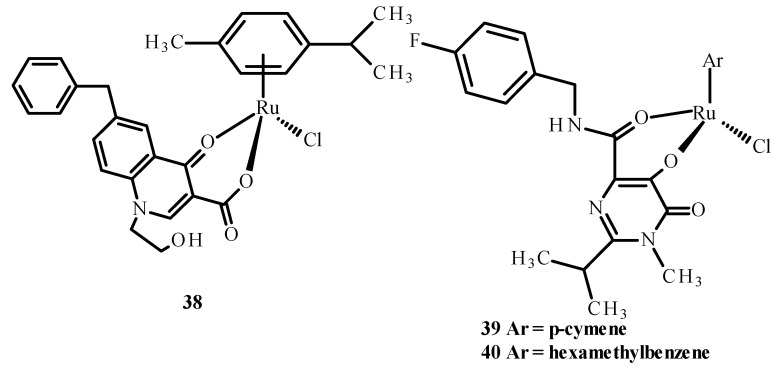
Structures of Ru(II) complexes-**38**, **39** and **40** [73].

**Figure 16 ijms-24-09512-f016:**
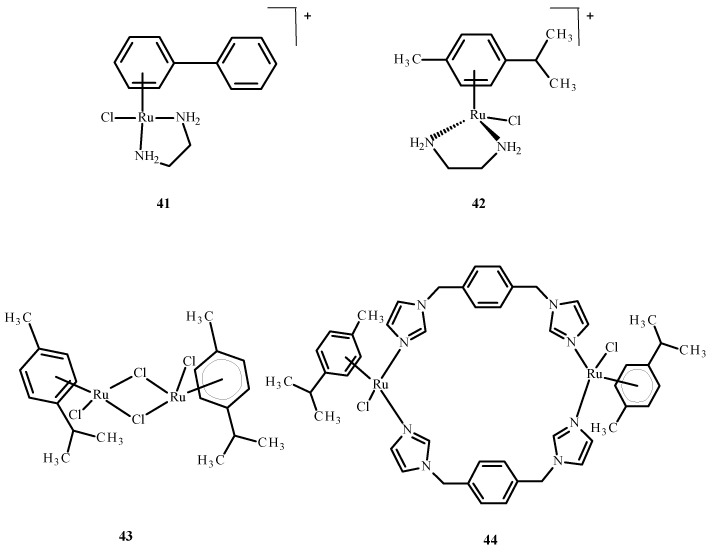
Ru(II) complexes **41**–**44** [73].

**Figure 17 ijms-24-09512-f017:**
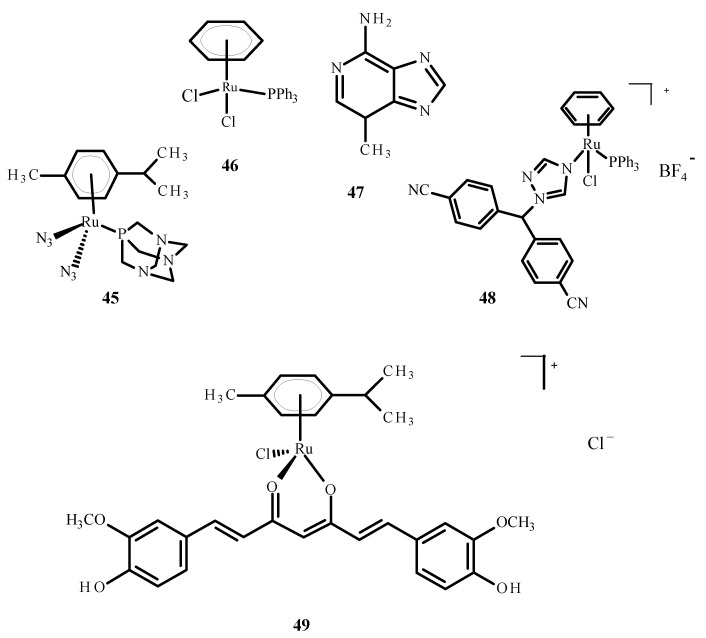
Structures of the arene Ru(II) complexes-**45**; **46** [40]; **48** [40]; **49** [79] and 3-methyladenine (**47**).

**Figure 20 ijms-24-09512-f020:**
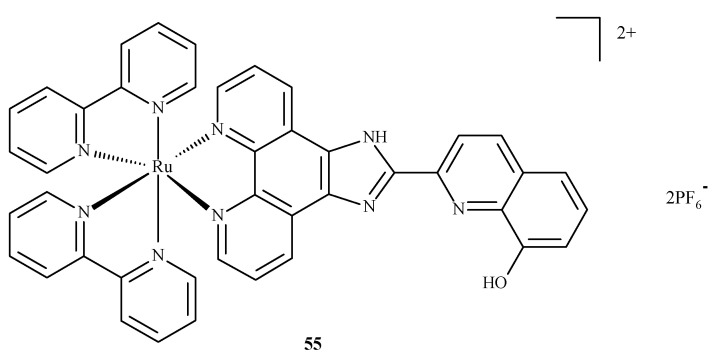
Structure of Ru(II) complex against Alzheimer’s disease **55** [85].

**Figure 21 ijms-24-09512-f021:**
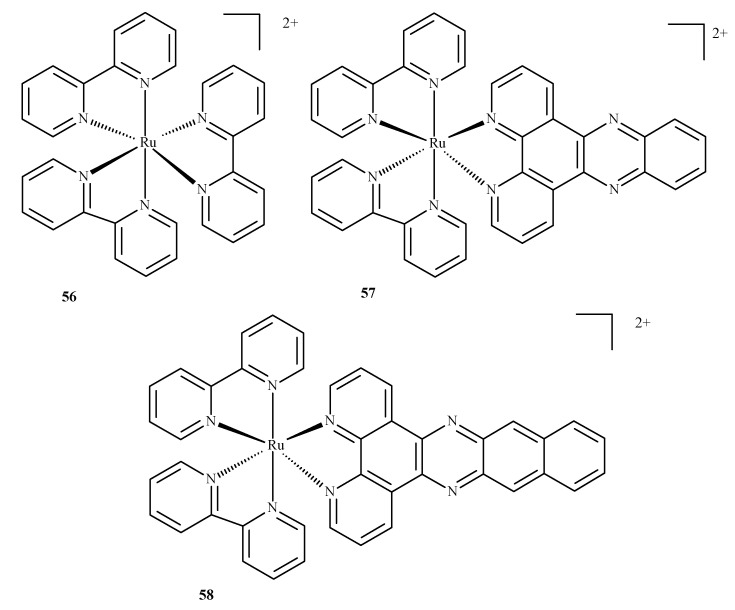
Complexes of Ru(II) with polypyridyl ligands **56**–**58** [87].

**Figure 22 ijms-24-09512-f022:**
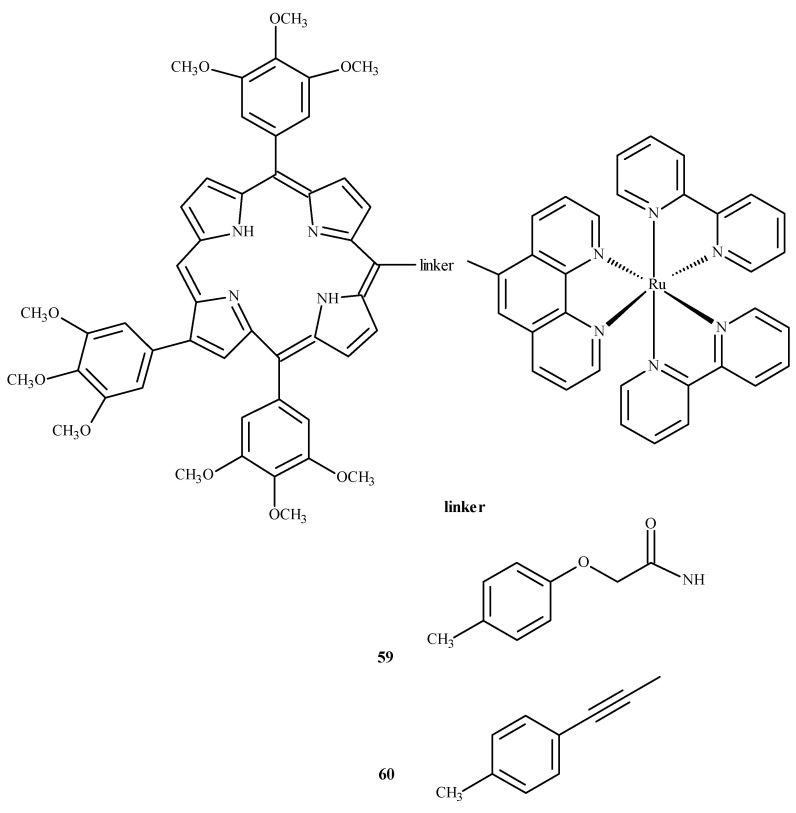
Complexes of Ru(II) with porphyrin-**59** and **60** [87].

**Figure 23 ijms-24-09512-f023:**
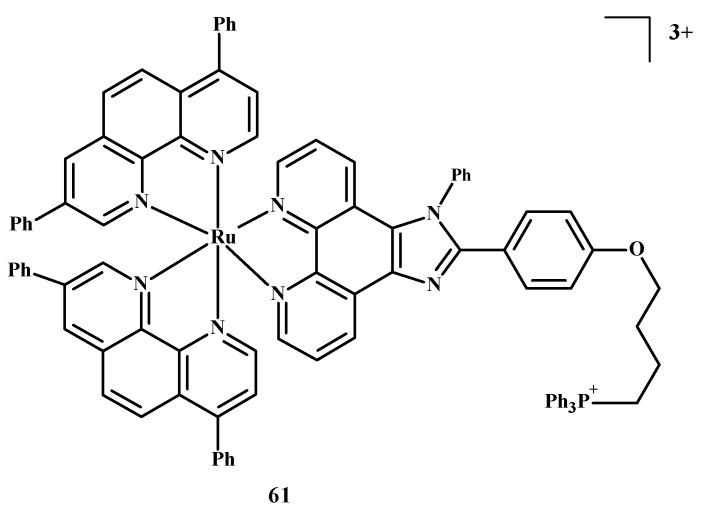
Structure of Ru(II) complex-**61** [10].

**Figure 24 ijms-24-09512-f024:**
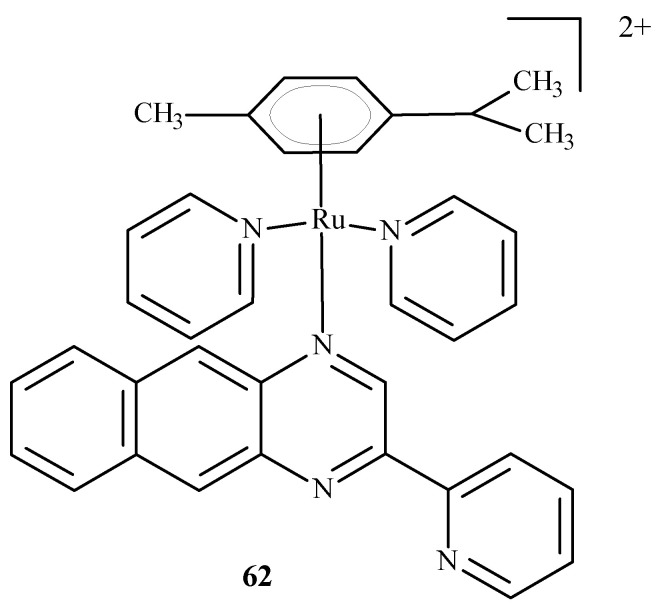
Structure of Ru[(p-cymene)Ru(dpb)(py)]^2+^ (**62**) [10].

**Figure 25 ijms-24-09512-f025:**
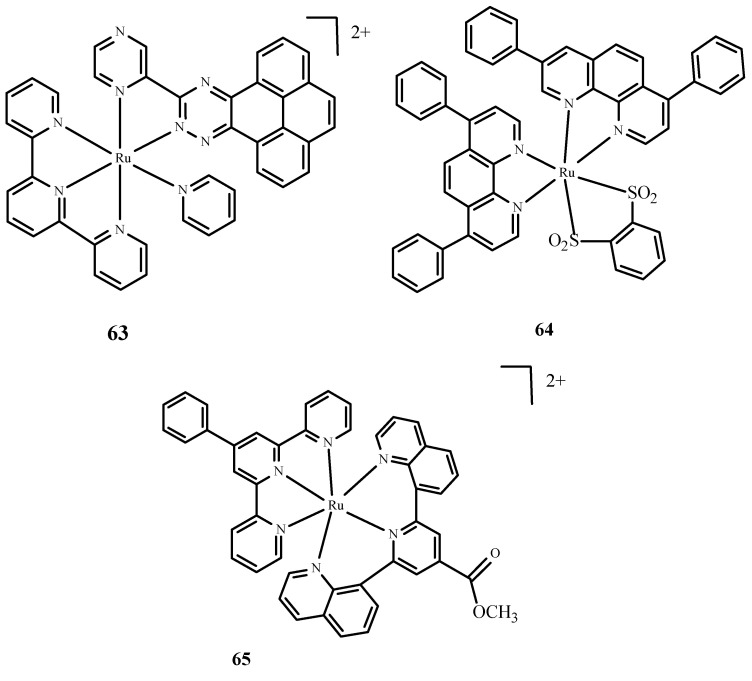
Complexes of Ru(II) **63**–**65** [10].

**Figure 26 ijms-24-09512-f026:**
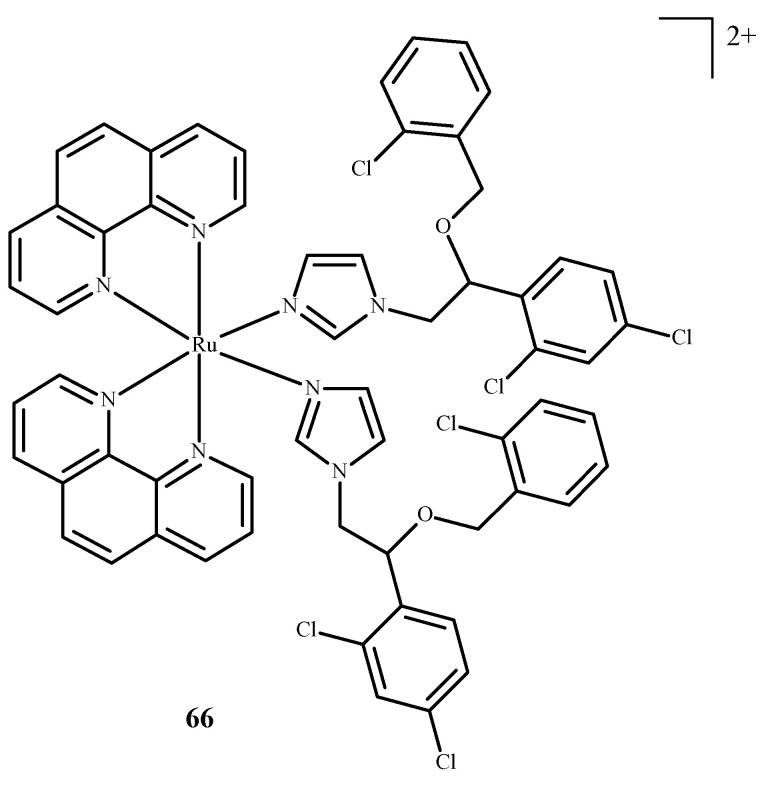
Complex of Ru(II) with econazole-**66** [10].

**Figure 27 ijms-24-09512-f027:**
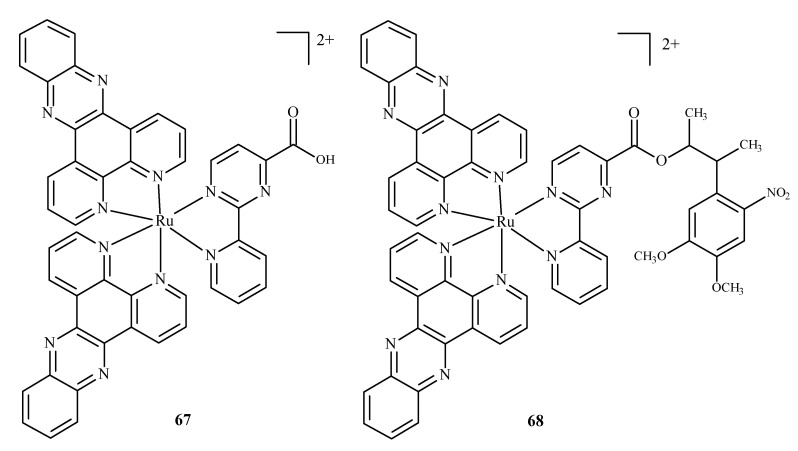
Complexes of Ru(II)-**67**, **68** [10].

**Figure 28 ijms-24-09512-f028:**
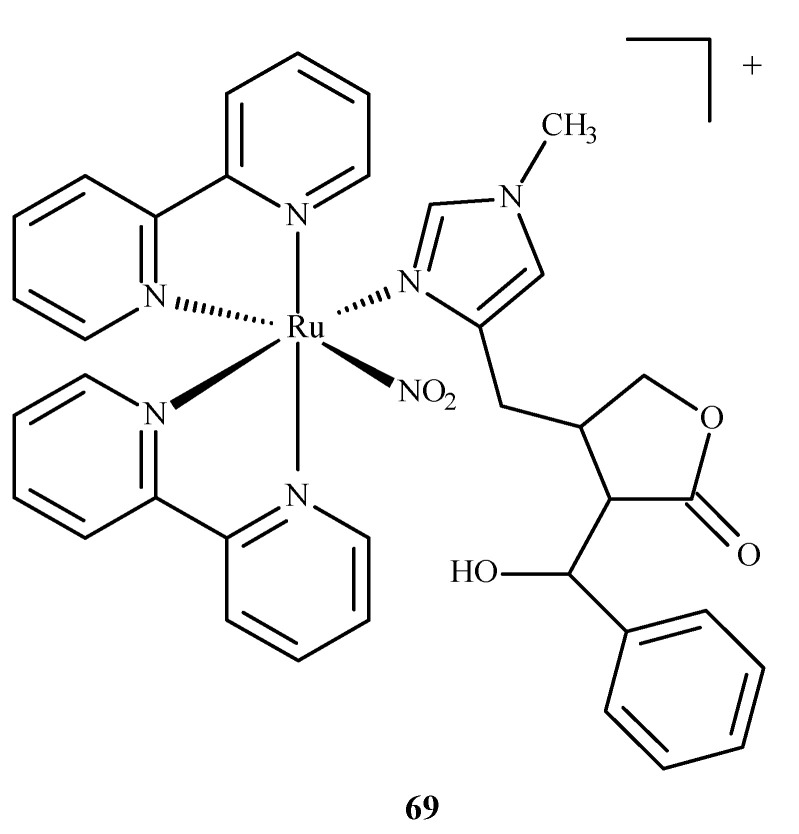
The complex [Ru(bpy)_2_(EPI)NO_2_]^+^, where bpy = 2,2′-bipyridine, studied as an anti-leishmaniasis agent.

**Figure 29 ijms-24-09512-f029:**
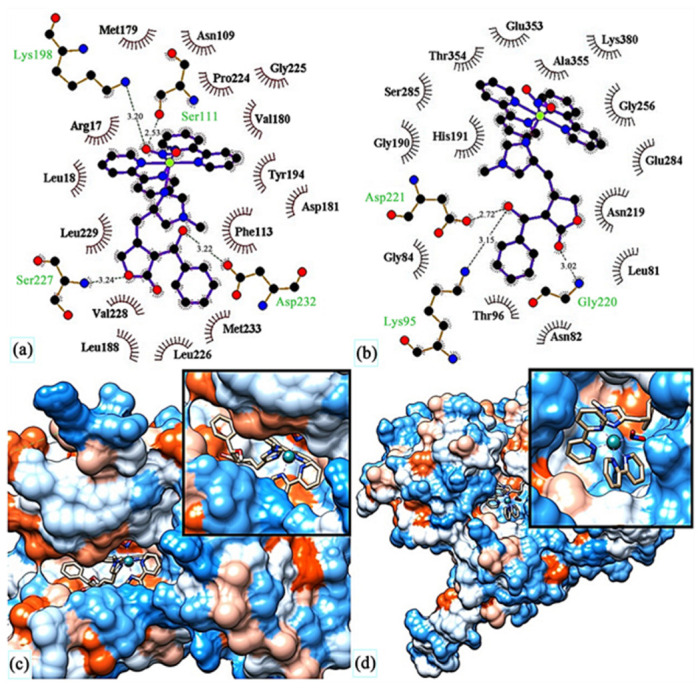
Molecular docking of (**69**) in (**c**) active site of 1e7w, (**d**) active site of 5nzg. The top images show molecular interactions between (**69**) and (**a**) 1e7w, (**b**) 5nzg. Reprinted with permission from Ref. [127]. Copyright 2023 by authors and Scientific Research Publishing Inc.

**Figure 30 ijms-24-09512-f030:**
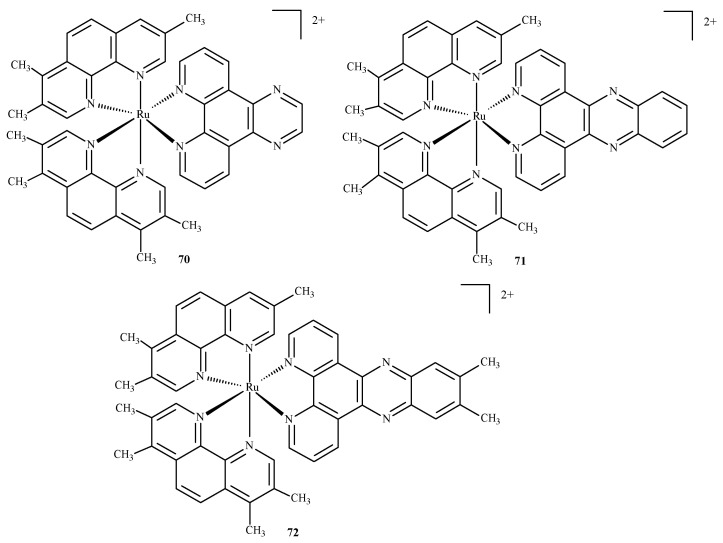
Three ruthenium(II) complexes studied by Das and Mondal [128].

**Figure 31 ijms-24-09512-f031:**
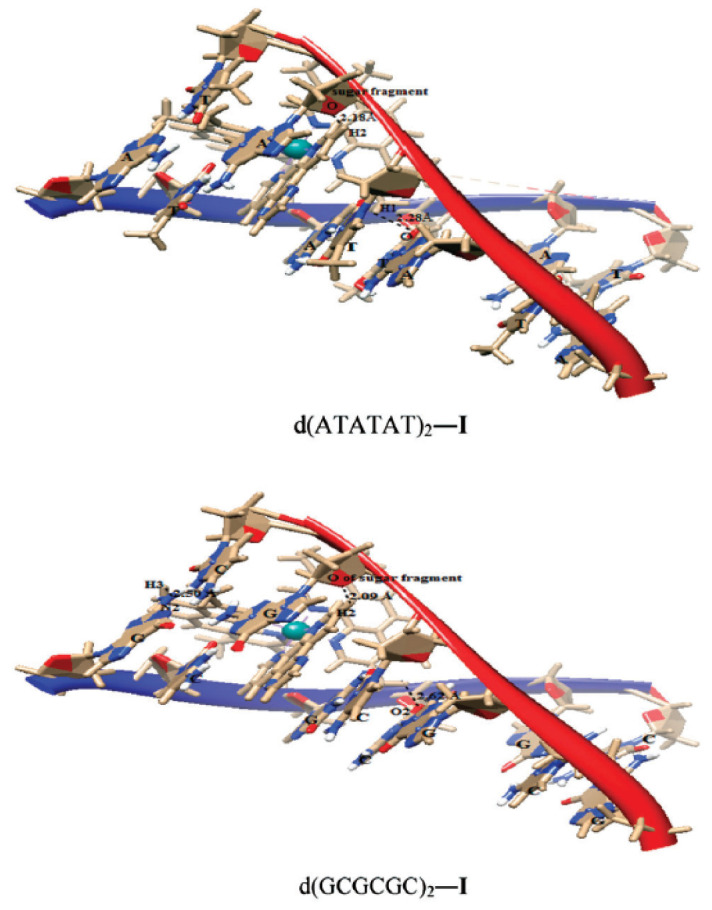
Complex **70** docked in d(ATATAT)_2_ and d(GCGCGC)_2_ sequences. Reprinted with permission from [128]. Copyright 2023 by the Centre National de la Recherche Scientifique (CNRS) and the Royal Society of Chemistry.

**Figure 32 ijms-24-09512-f032:**
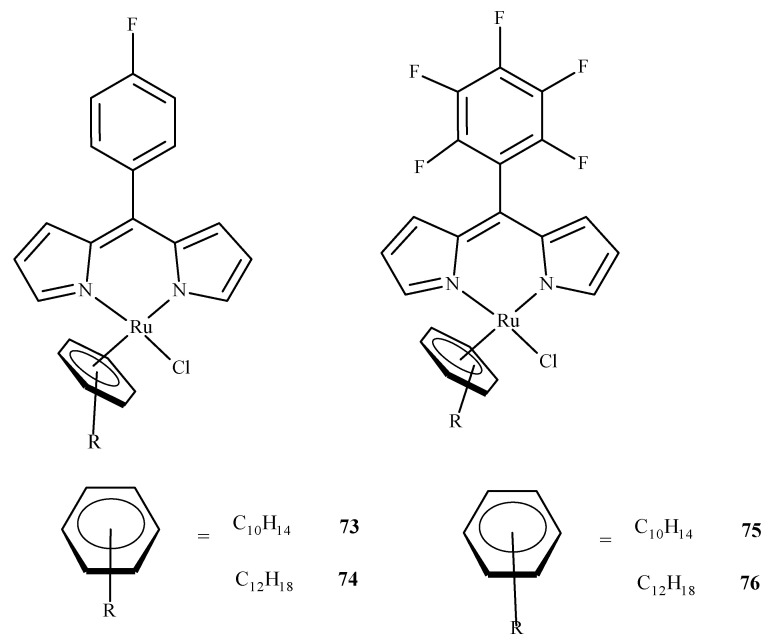
Four dipyrrin-based ruthenium(II) complexes.

**Figure 33 ijms-24-09512-f033:**
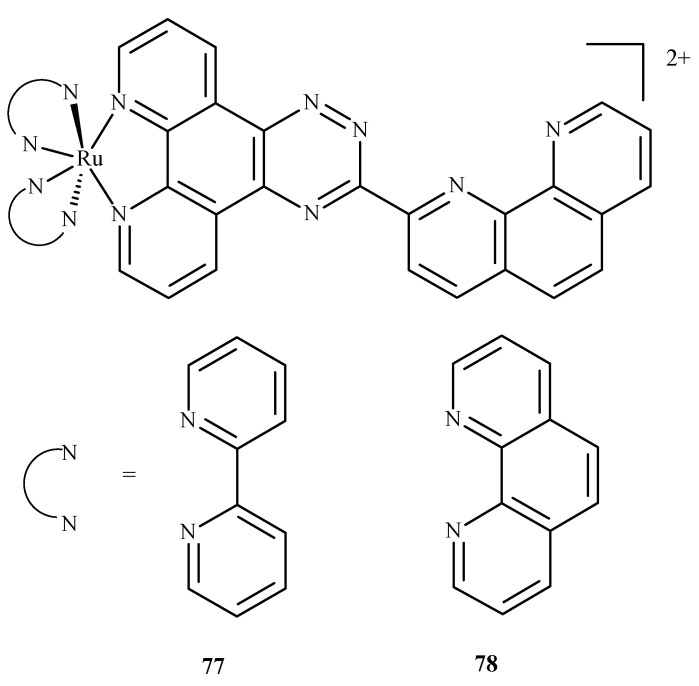
Compound (**77**) is [Ru(bpy)_2_(ptpn)]^2+^, and (**78**) is [Ru(phen)_2_(ptpn)]^2+^. Both were studied as stabilizing agents of the human telomeric quadruplex.

**Figure 34 ijms-24-09512-f034:**
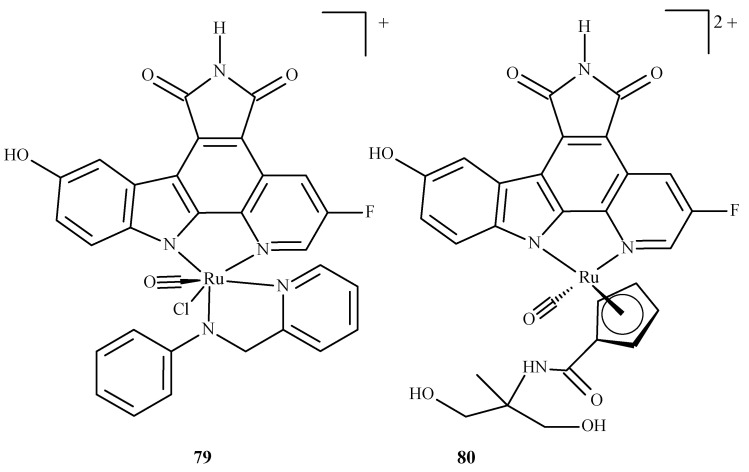
Ruthenium(II) complexes as potential inhibitors of PfCDPK2 considered in the study by Patel et al. [147]: **79** being Ru-pyridocarbazole (also known as FLL) **80** being methylated Ru-pyridocarbazole (E52).

**Figure 35 ijms-24-09512-f035:**
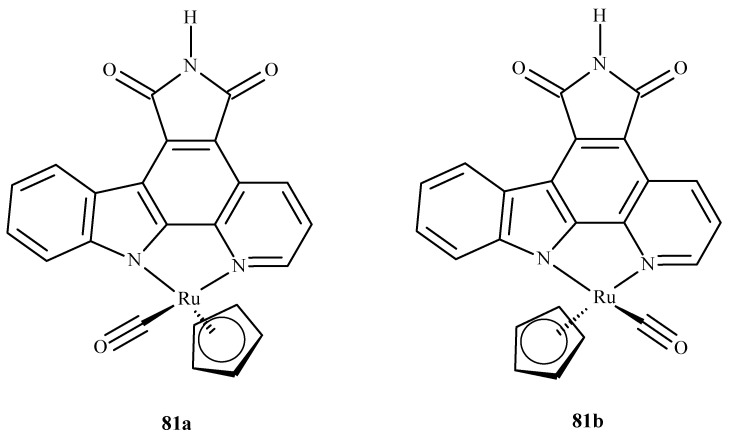
Two half-sandwich complexes of ruthenium, (**left**) being the R enantiomer and (**right**) the S enantiomer.

**Figure 36 ijms-24-09512-f036:**
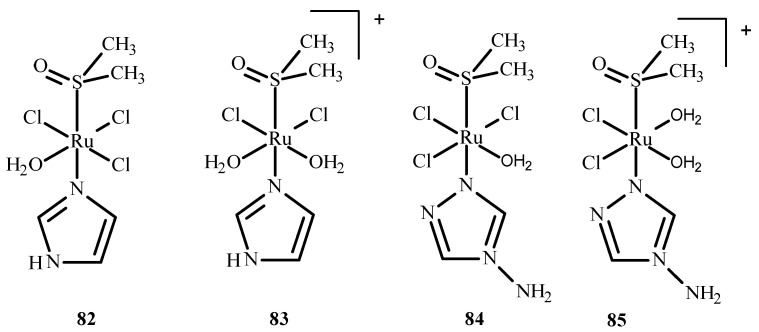
Four ruthenium(III) complexes, (**82**) and (**84**) being monoaqua and (**83**) and (**85**) being diaqua.

**Figure 37 ijms-24-09512-f037:**
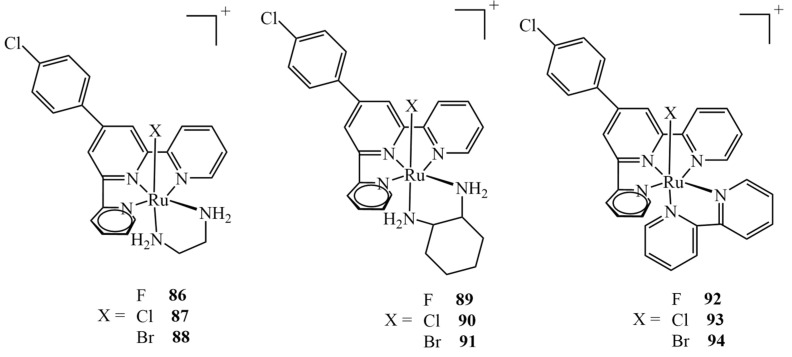
Nine ruthenium(II) complexes investigated theoretically as potential anticancer agents, out of which only chloro-derivatives **87**, **90** and **93** were studied experimentally.

## Data Availability

No data were used for the research described in the article.

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
