# Peer review of "An Overview of the Potential Medicinal and Pharmaceutical Properties of Ru(II)/(III) Complexes"

_ijms, 2023, doi:10.3390/ijms24119512_

Round 1

Reviewer 1 Report

The manuscript proposed by Skoczynska and co-workers for publication in Inter. J. Molecular Sciences entitled «An overview of potential medicinal and pharmaceutical properties of Ru(II)/(III) complexes» in its current form is not suitable for publication and should be considered for major revision. Below are comments which should help to improve the manuscript:

·       My main concern is that there are several recent reviews on this topic; some examples: 1) Ruthenium Metallotherapeutics: Novel Approaches to Combatting Parasitic Infections (2022) (doi: 10.2174/0929867329666220401105444); 2) Ruthenium Complexes as Anticancer Agents: A Brief History and Perspectives (2020) (doi: 10.2147/DDDT.S275007); 3) Toward Multi-Targeted Platinum and Ruthenium Drugs—A New Paradigm in Cancer Drug Treatment Regimens? (2019) (doi: 10.1021/acs.chemrev.8b00271); 4) Designing Ruthenium Anticancer Drugs: What Have We Learnt from the Key Drug Candidates? (2019) (doi: 10.3390/inorganics7030031); 5) Applications of Ruthenium Complex in Tumor Diagnosis and Therapy (2018) (doi: 10.3389/fphar.2018.01323); 6) Ruthenium coordination compounds of biological and biomedical significance. DNA binding agents (2018) (doi: 10.1016/j.ccr.2018.07.012); 7) Structure–activity relationships for ruthenium and osmium anticancer agents – towards clinical development   (2018)  (doi: 10.1039/C7CS00332C) These papers should also be cited. What is the novelty of the present manuscript?

·       Generally speaking, the manuscript feels like it is not finished yet. There is a huge room for improvement; much more could have been done to improve this work. Authors should keep in mind the review paper style, and the available data should be adequately demonstrated and discussed (critics, evaluation, not just writing the literature results) briefly. At least the authors should insert a brief summary after each subchapter.

·       In some parts, the text is hard to read. Example: lines 97-100, 

·       There are throughout the text many abbreviations without meaning.

·       Rewrite the sentence: “The ability of Ru(II)/(III) complexes to bind to DNA varies depending on the cytotoxicity” (lines 77-78). It is incorrect!

·        

·       The chemical structures were not drawn in the same style (Fig 56-58, 61, 69, etc.)

·       The discussion of mechanisms of action of Ru(II)/(III) is somewhat limited. The wording of this part could be greatly improved.

·       Figure 19 is unnecessary.

·       Have the authors’ copyright permission to reproduce Fig. 29 and 31?

·       Add the reference to Fig 30.

·       In the conclusion, I’m afraid I have to disagree with the sentence “This review article refers to the potential application of ruthenium complexes in the medical and pharmaceutical profile” (763-764) because, throughout the manuscript, the authors highlighted the only pharmaceutic potential of Ru compounds. For the same reason, I also disagree with the title of the manuscript.

·       Throughout the manuscript, the authors highlight that the Pt(II) complexes have a problem with side effects. However, resistance development is the main issue related to Pt(II) complexes, which makes these compounds ineffective.

·       The verb imitate in the article needs to be replaced by mimic.

·        I think the name of section 3.4. (Cytotoxic complexes of Ru(II)/(III)) is not adequate. Authors need to continue the same pattern of the other sections. In addition, I think that section 2 (Various mechanisms of action of Ru(II)/(III) complexes and their therapeutic targets) could be included in section 3.4, because the authors just write about the antitumor action mechanism.

·       The discussion about Ru compounds’ ability to strongly bind to nucleic acids and protein is poor in section 3.4.

·       The design of Fig 2 could be improved.

·       In the introduction, the author could approach more about the advantages of Ruthenium-containing compounds that are the most promising non-platinum candidates for metal-based cancer therapy due to their suitable features for anticancer drug design. For example, the authors just mentioned the possibility of reduction and octahedral geometry about Pt complexes. Still, the ligand exchange kinetics similar to platinum compounds and preferential accumulation in neoplastic tissues, the authors didn’t mention.

·       In addition, three organoruthenium compounds have reached the clinical evaluation stage in humans, and the authors didn’t mention it. 

Some parts of the text are hard to read, section 2, for instance. 

Author Response

Responses to the Reviewer 1

The authors thank the reviewer for very useful suggestions and we hope that  our corrections will be acceptable.

Comment 1: In some parts, the text is hard to read. Example: lines 97-100

Response 1: We have rewritten the text to make it more readable in English.

‘Studies of [Ru(η6-C6H6)(DMSO)Cl2] (4), the first arene Ru(II) compound (Figure 3), revealed that it strongly inhibits topoisomerase II, which participates in DNA replication and is responsible for structural organization [21].’

Comment 2: There are throughout the text many abbreviations without meaning.

Response 2: We have explained abbreviations in the text, where they appear for the first time.

Comment 3: Rewrite the sentence: “The ability of Ru(II)/(III) complexes to bind to DNA varies depending on the cytotoxicity” (lines 77-78). It is incorrect!

Response 3: We have corrected:

‘The ability of ruthenium complexes to interact with DNA varies and this is reflected in their cytotoxicity: some compounds of Ru(III) interact with nuclear DNA, decrease RNA synthesis, inhibit DNA replication, have mutagenic activity.’

Comment 4: The chemical structures were not drawn in the same style (Fig 56-58, 61, 69, etc.)

Response 4: We have drawn all structures in the same style.

Comment 5: The discussion of mechanisms of action of Ru(II)/(III) is somewhat limited. The wording of this part could be greatly improved.

Response 5:We do not think the discussion of mechanisms is limited, because we wanted to focus on the most important issues connected with this topic.

Comment 6: Figure 19 is unnecessary.

Response 6: We do not think, that Figure 19 is unnecessary, because it is an interesting example of DNA damaging activity by Ru(II) complex.

Comment 7: Have the authors’ copyright permission to reproduce Fig. 29 and 31?

Response 7: The reference 127 containing Figure 29 has been published under CC BY 4.0 licence and does not require copyright permission. The permision for reproducing Figure 31 from the reference 128 has been obtained.

Comment 8: Add the reference to Fig 30.

Response 8: It has been done.

Comment 9: In the conclusion, I’m afraid I have to disagree with the sentence “This review article refers to the potential application of ruthenium complexes in the medical and pharmaceutical profile” (763-764) because, throughout the manuscript, the authors highlighted the only pharmaceutic potential of Ru compounds. For the same reason, I also disagree with the title of the manuscript.

Response 9:We have thought about this comment and decided not to change the title, because we described here application of Ru(II)/(III) complexes in clinical trials, which are performed in humans. That proves a seriously application of described complexes as medicines.

Comment 10:  Throughout the manuscript, the authors highlight that the Pt(II) complexes have a problem with side effects. However, resistance development is the main issue related to Pt(II) complexes, which makes these compounds ineffective.

Response 10: We agree with that finding. We have mentioned about problem with resistance of Pt(II) complexes in Section 1, Subsection 3.1.

Comment 11:  The verb imitate in the article needs to be replaced by mimic.

Response 11: We have replaced the verb imitate with mimic.

Comment 12: I think the name of section 3.4. (Cytotoxic complexes of Ru(II)/(III)) is not adequate. Authors need to continue the same pattern of the other sections. In addition, I think that section 2 (Various mechanisms of action of Ru(II)/(III) complexes and their therapeutic targets) could be included in section 3.4, because the authors just write about the antitumor action mechanism.

Response 12: We have changed the title of subsection 3.4. to ‘Cytotoxicity of Ru(II)/(III) complexes’. We realize that Section  2 and Subsection 3.4 are related, but we have not combined them, because chapter 2 contains a separate issue, which is important to understand the cytotoxicity of complexes.

Comment 13: The discussion about Ru compounds’ ability to strongly bind to nucleic acids and protein is poor in section 3.4.

Response 13: Section 5 discusses interactions between complexes and biomolecules and illustrates examples of such interactions.

Comment 14: The design of Fig 2 could be improved.

Response 14: We have changed and modified the Figure 2 and think that now is more legible and clear to understand the mechanisms of action of Ru(II)/(III) complexes.

Comment 15:  In the introduction, the author could approach more about the advantages of Ruthenium-containing compounds that are the most promising non-platinum candidates for metal-based cancer therapy due to their suitable features for anticancer drug design. For example, the authors just mentioned the possibility of reduction and octahedral geometry about Pt complexes. Still, the ligand exchange kinetics similar to platinum compounds and preferential accumulation in neoplastic tissues, the authors didn’t mention.

Response 15: We have modified the introduction, and we have tried explained the advantages of Ru(II)/(III) complexes in comparison to Pt(II) complexes. We have paid more attention to ruthenium complexes, as they are the subject of this manuscript.

Comment 16: In addition, three organoruthenium compounds have reached the clinical evaluation stage in humans, and the authors didn’t mention it. 

Response 16: We have described complexes of Ru(II)/(III) which have reached the stage of clinical trials in the Subsection 3.1. We have mentioned about accumulation in the neoplastic cells and a ligand exchange kinetics.

Reviewer 2 Report

Comments and suggestions for Authors are collected in the attached file.

In my opinion, the manuscript mostly was prepared with correct language, but there are passages that require minor improvements (details included in the attached file with the review report).

Author Response

Responses to the Reviewer 2

The authors thank for very useful comments for our manuscript.

Comment 1:

The manuscript prepared by Skoczynska et al. deals with a very interesting issue, which is the potential use of Ru(II)/(III) complexes in medicine and pharmaceutics. The interest in such compounds stems from the ongoing search for replacements for platinum compounds such as cisplatin, carboplatin, or oxaliplatin, which, although already used in oncology, carry numerous and very troublesome side effects for patients. I believe that the chosen topic is interesting and certainly is of great interest to the scientific world, but the manuscript itself needs some improvements and corrections before publication. The manuscript under review is based on 150 items of literature, the vast majority of which (110) cover the last 10 years, with 53 items being literature from the last 5 years. However, there are papers which have been published this year (e.g., https://doi.org/10.1016/j.matpr.2022.07.098) and therefore the literature list should be updated.

Response 1:

In this manuscript, we used publications, most of which are from the period 2012-2022, while a few references are from the period 2000-2010. A lot of information is duplicated in the cited publications, which, despite their recent appearance, are based on older publications.

Comment 2:

I think a little more space in the manuscript should be devoted to the possible mechanisms involved in the anticancer activities of Ru complexes (at the moment there is only a diagram shown as Figure 2). Perhaps it would be worthwhile to present the individual mechanisms of action in a little bit more extensive way. The Authors devoted very little space to ruthenium complexes acting as radiosensitizers or photothermal therapy in cancer treatment. It seems to me that these aspects should have been mentioned in the manuscript. Moreover, the Authors completely omitted the issue of nanocarriers used as supporting elements for the distribution of Ru complexes to planned targets. I believe that a short section should also be devoted to this topic, which has been gaining more and more attention among researchers in recent years (e.g., https://doi.org/10.1038/s41570-019-0088-0; https://doi.org/10.1021/acsabm.1c00151; https://doi.org/10.3390/pharmaceutics13040460 , etc.)

Response 2: The authors thank the reviewer for critical comments, but we believe that it is impossible to include all the information regarding the role of ruthenium ions in such an extensive article. Nevertheless, due to the fact that other publications on this subject are planned, we will certainly take this suggestion into account. We corrected  the Figure 2.

Comment 3: Introduction. In my opinion, this part of the paper is written rather chaotically and does not fulfill its role as an introduction to the rest of the manuscript. The Authors mention cisplatin at the beginning (lines 34- 41), then they describe the properties of Ru and its complexes, only to return again to background information regarding cisplatin (66-67). In addition, some sentences seem unclear, such as:

Lines 40-44 – “Their other side effects are loss of hair, nausea, suppression of bone marrow, vomiting and resistance to of metallopharmaceuticals in the treatment of cancer [1,2,3,4], because they have six-coordinated octahedron geometry and different degrees of oxidation [1,10,5,6], which make them more attractive than Pt(II) analogues and allow easier ligand modifications”. Here, the beginning of the sentence refers to Pt complexes, but the next part already contradicts the beginning of the sentence, just because cisplatin and its derivatives labeled in the paper as (2) (3) do not have "six-coordinated octahedron geometry"....

Response 3: We have corrected the mistakes as below:

‘Despite their anticancer effectiveness, they have some disadvantages, such as toxicity to the excretory and nervous systems, as well as loss of hair, nausea, suppression of bone marrow, vomiting and resistance to drugs [1,2,3,4,5].

As many therapies fail due to the development of resistance of tumours to platinum compounds, there is great interest in identifying alternative anticancer drugs with mild side effects. Much hope has been placed on complexes of Ru(II)/(III) ions, which are candidates for the next generation of metallopharmaceuticals in the treatment of cancer  [1,6,7,8], because they have six-coordinated octahedron geometry and different degrees of oxidation drugs [1,2,9,10], which make them more attractive than Pt(II) analogues and allow easier ligand modifications.’

Lines 47-48 – “This supports the selectivity of ruthenium complexes when entering neoplastic cells [1,10,7,8,9]. drugs [1,10,11,12,13].” This sentence in this form is incomprehensible to me.

We have corrected the sentence as below:

‘The low kinetics of ligand exchange enables the attachment of Ru(II/III) complexes to certain cell structures throughout the cell cycle. Ru(III) ions can expoit the transferrin overexpression in the cancer cells by mimicing Fe(III) ions and binding to the protein. This supports the selectivity of ruthenium complexes when entering neoplastic cells [1,2,11,12,13].’

Subsection 3.1.

Comment 4: Lines 182-183 – “RAPTA-C (5) (Figure 6)” – this compound is not present in Figure 6 and, consequently, it is not clear to which Ru complex the information given in this sentence refers.

Response 4: RAPTA-C is in the Figure 3 and we have corrected the mistake.

Subsection 3.3.

Comment 5: Line 266 – unfinished sentence: “Eilatin is an alkaloid derived from...”.

Response 5: We have finished the sentence: Eilatin is an alkaloid derived from the marine tunicate Eudiastoma sp.

Comment 6: Lines 266-268 – “The anti-HIV activity against HeLa cells infected with HIV-1 was 0,8 μM...” – the Authors should specify how this anti-HIV activity was determined, i.e. what is expressed by the given value of 0.8 μM (the optimal concentration of the test substance, its IC50, or maybe some other parameter).

Response 6: Value 0,8 µM refers to IC50, which is concentration needed to decrease HIV-1 activity by 50% in a plaque-formation assay.

Comment 7: Lines 273-276 – the Authors provide IC50 values, but do not relate them to any reference values, so it is difficult to conclude from this short passage about the usefulness of the studies described in the literature items [71-73]. I believe that this section of the text should be supplemented with additional information.

Response 7: The original publication (Luedtke, N.W.; Hwang, J.S.; Glazer, E.C.; Gut, D.; Kol, M.; Tor, Y. Eilatin Ru(II) complexes display anti-HIV activity and enantiomeric diversity in the binding of RNA. Chembiochem. 2002, 3, 766–771) presents IC50 values for structures and their racemic mixtures, but there is no comparison of these results with reference values.

Subsection 3.4.

Comment 8: Line 328 – “combination with erlotinib (Figure 3)” – are you sure that the structure of the compound named erlotinib is shown in Figure 3?

Response 8: Figure 3 refers to complex RAPTA-C, not to the complex RAPTA-C with erlotinib.

Comment 9: Line 350 – „Complex 50 (Figure 17)” – there is no structure 50 in Figure 17...

Response 9: We have corrected the number 50 and replaced it with 49.

Comment 10: Line 356 – the SD value for IC50 is not given (the same is true for other IC50 values discussed further in the text).

Response 10: We have corrected and the value should be 36±6 µM.

Comment 11: Line 358 – the notation form used suggests that the IC50 values described are presented in Figure 17.

Response 11: We have removed ‘Figure 17’.

Comment 12: Line 359 – I do not understand this form of writing: ”was 11.3±1.4 μM (48) [78].” It gives a wrong impression that this IC50 value is somehow shown together with the structure (48)....

Response 12: We have changed it.

Comment 13: Line 371 – it should be explained what “COLO-205” stands for.

Response 13: COLO 205 means Human Colon Adenocarcinoma

Subsection 3.5.

Comment 14: Lines 390-398 and 398-405 – they give the same information with almost the same phrases. However, there is a puzzling discrepancy between the numbers of complexes discussed: (56) in line 392 and (55) in line 399...

Response 14: We have changed 56 to 55.

Section 4

Comment 15: Lines 439-440 – several ligands are listed, but their structures are not shown (the corresponding figures were prepared for other complexes discussed in the manuscript). It might be worthwhile to show their structures...

Response 15: We do not think, that the structures of ligands are necessary, because, they are included in the complexes discussed in the Section 4.

Comment 16: Lines 443-445 and 452-454 – repetition of the same information.

Response 16: We have removed the sentence in lines below:

‘The phototoxic effect varied depending on the used isomer, with three-substituted systems having stronger phototoxic properties than the four-substituted analogs.’

Comment 17: Figure 21 – there is no reference in the text to the structure marked (58).

Response 17: The reference [84] is cited in the end of the Section 4.

  1. Pobłocki, K.; Drzezdzon,J.; Kostrzewa, T.; Jacewicz, D. Coordination Complexes as a New Generation Photosensitizer for Photodynamic Anticancer Therapy. Int. J. Mol. Sci. 2021, 22, 1-16.

Comment 18: Figure 22 – what is „linkier”? Did you mean “linker”?

Response 18: Yes, there should be ‘linker’.

Comment 19: Lines 461-471 – this passage sounds more like an introduction, so I guess it should be at the beginning of this section.

Response 19: We do not think that lines 461-471 sounds like an introduction. It is at the beginning of a new paragraph and should be treated as a reminder.

Section 5

Comment 20: Lines 709 and 715 – it should be (81b) instead of (82b).

Response 20: We have corrected it.

Comment 21: Figure 36 – in the caption, there are (a)-(d) notations which are not present in the figure itself.

Response 21: We have corrected it.

Comment 22: Line 752 – it should be (93) instead of (03)

Response 22: We have corrected it.

Comment 23: In general:Abbreviations that are not explained often appear in the paper, e.g. PARP, EGFR, HUVEC, TNBC, MTTin lines 154, 155, 331, 341, and 355, respectively. Authors should add a list of abbreviations used orexplain them in the text when they appear for the first time.

Response 23: We have explained abbreviations in the text, where they appear for the first time.

Comment 24: In my opinion, the work needs some linguistic correction, especially chapters 1-4 and 6 contain sentences that are not fully understood, such as:

Response 24: We have performed linguistic corrections in the chapters 1-4 and 6 and corrected sentences in lines 151, 153, 321, 318, 776.

Comment 25: Line 151: „The mechanism of action of relies on is accumulation in the mitochondria”

Response 25: We have corrected and replaced ‘is’ with ‘its’.

Comment 26: Line 153:

Response 26: The sentence is: ‘The mechanism of action of 16 relies on its accumulation in the mitochondria and activation of superoxide dismutase (SOD).’

Comment 27: Line 321: „of their used in diagnostics and therapies” or words used in the wrong way:

Response 27: The sentence is: ‘A lot of metal ion complexes have significantly influenced the development of anti-tumour drugs, because of their use in diagnostics and therapy [1].’

Comment 28: Lines 318 and 776: „The last one resist on blocking...” and „...activity mechanisms of ruthenium complexes resist on the epigenetic scheme...”

Response 28:

We have corrected: ‘Furthermore, ruthenium complexes are demonstrate epigenetic activities, e.g. interaction with biomolecules, including coordination to the nucleosome core, formation of adducts with histones and inhibition of topoisomerase II.’

Comment 29: Also unintelligible are the sentences in lines 273-276: „The other complexes 34 and 35 (...) Other complex 34...” and 450-451: „The test performed at 652 nm in human melanoma cells (Me300), that the compounds at a concentration of 10 μM had 60-80% cytotoxicity”.

Response 29: We have corrected the sentences as below:

‘The other complexes 34 and 35 (Figure 13) were analysed as potential inhibitors of HIV-1 and HIV-2 virus in human T Cell leukaemia (MT-4). Complex 34 (Figure 13) showed an IC50 value higher than 0.21 µM, while the IC50 of complex 35 (Figure 13) was higher than 2.14 µM.’

‘At a concentration of 10 µM, the compounds demonstrated 60-80% cytotoxicity against human melanoma cells (Me300), measured at 652 nm.’

Leaving aside all the above-mentioned comments, the manuscript also needs work on text editing:

Comment 30: double [e.g., lines 60, 162...] or unnecessary spaces [e.g., lines 448, 673....] appear quite frequently;

Response 30: We have removed unnecessary spaces.

Comment 31: numerical values from SD are written differently [e.g., lines 358 and 359, 370 and 371]

Response 31: We have removed spaces between ‘±’ and SD.

Comment 32: decimal places in numbers should be marked with a period, not a comma [e.g., lines 267, 490];

Response 32: We have corrected it and used periods in decimal places.

Comment 33: the structure (20) shown in Figure 6 lacks subscripts;

Response 33: Structure (20) which is in the Figure 6 has subscripts. It is minimized, because Figure 6 contains 6 quite large structures.

Comment 34: the structure labeled (17) appears twice in different figures, i.e. Figure 6 and Figure 16. It seems redundant to me;

Response 34: We have removed structure 17 from the Fig. 16.

Comment 35: the Authors should standardize the notation of the numbering of structures in the paper, namely with or without brackets [e.g., lines 152 and 153, 241 and 245, etc.];

Resposne 35: We have used numbering in brackets, when the structure name was written.

Comment 36: I wonder about the necessity of adding in the text each time information about the figure number in which the structure is shown – the structures themselves already have a numbering and it is used to identify the compounds discus

Resposne 36: We have added figure numbers, because we wanted to make this manuscript easier to read.

Round 2

Reviewer 1 Report

I have just some suggestions:

·       Section 2: Various mechanisms of action of Ru(II)/(III) complexes and their therapeutic targets. The word “various” could be replaced by “Main.”

·       Section 3: Medicinal and pharmaceutical side of Ru(II)/(III) complexes. I suggest using the word “potential” or “applications” instead of “side.”

·       Rewrite section 3.5’s title: “Anti -Alzheimer's disease complexes of Ru(II)/(III).”

·    Section 5: “Computational approaches to interactions of Ru(II)/Ru(III) complexes with their biological targets”. I missed one verb after “approaches to…” investigate/study/ explore/comprehend, etc…

The English could be improved. In the comments for authors, I suggested some modifications to the section's title, but there are other Eng. mistakes throughout the text. 

Author Response

Responses to the Reviewer 1

Comment 1:  Section 2: Various mechanisms of action of Ru(II)/(III) complexes and their therapeutic targets. The word “various” could be replaced by “Main.”

Response 1: We have replaced the word “various” with “main”.

Comment 2: Section 3: Medicinal and pharmaceutical side of Ru(II)/(III) complexes. I suggest using the word “potential” or “applications” instead of “side.”

Response 2: We have replaced the word “side” with “potential”.

Comment 3:  Rewrite section 3.5’s title: “Anti - Alzheimer's disease complexes of Ru(II)/(III).”

Response 3: We have rewritten the title of section 3.5 as below:

“Ru(II)/(III) complexes with potential anti Alzheimer’s disease properties”

Comment 4: Section 5: “Computational approaches to interactions of Ru(II)/Ru(III) complexes with their biological targets”. I missed one verb after “approaches to…” investigate/study/ explore/comprehend, etc…

Response 4: We have added ‘studying’ and the sentence is:

“Computational approaches to studying interactions of Ru(II)/Ru(III) complexes with their biological targets”

Reviewer 2 Report

The Authors have addressed most of the comments. I also accept their arguments in the polemic with a Reviewer. However, there is still one point that needs to be addressed before the manuscript is submitted for publication:

Comment 17: Figure 21 – there is no reference in the text to the structure marked (58).

Response 17: The reference [84] is cited in the end of the Section 4.

  1. Pobłocki, K.; Drzezdzon,J.; Kostrzewa, T.; Jacewicz, D. Coordination Complexes as a New Generation Photosensitizer for Photodynamic Anticancer Therapy. Int. J. Mol. Sci. 2021, 22, 1-16.

The Authors should refer somehow in the body text to the STRUCTURE (58). This structure is shown in Figure 21, but there is no reference to it in the text, which means that this complex is not discussed in the text in any way. In that case, why is it shown? What relevance does it have to the issues raised in the study?

Unlike the structure 58, I am able to find the reference to the literature item [84]. Actually, I really don't know why exactly this item is listed here, since in the review report I refer to the structure (58)…

Author Response

Response to the Reviewer 2

Comment 1: The Authors should refer somehow in the body text to the STRUCTURE (58). This structure is shown in Figure 21, but there is no reference to it in the text, which means that this complex is not discussed in the text in any way. In that case, why is it shown? What relevance does it have to the issues raised in the study?

Unlike the structure 58, I am able to find the reference to the literature item [84]. Actually, I really don't know why exactly this item is listed here, since in the review report I refer to the structure (58)…

Response 1: We have added the number 58 in the sentence below:

“Recently an investigation of differences between compounds 57 and 58 (which contains benzo[i]dipyrido [3,2-a:20,30-c]phenazine (dppn)) was conducted.”
